# SPACE-TIME GRAPH NEURAL NETWORKS

**Samar Hadou, Charilaos I. Kanatsoulis & Alejandro Ribeiro**
Department of Electrical and Systems Engineering
University of Pennsylvania
`{selaraby, kanac, aribeiro}@seas.upenn.edu`

## ABSTRACT

We introduce *space-time graph neural network (ST-GNN)*, a novel GNN architecture, tailored to jointly process the underlying space-time topology of time-varying network data. The cornerstone of our proposed architecture is the composition of time and graph convolutional filters followed by pointwise nonlinear activation functions. We introduce a generic definition of convolution operators that mimic the diffusion process of signals over its underlying support. On top of this definition, we propose space-time graph convolutions that are built upon a composition of time and graph shift operators. We prove that ST-GNNs with multivariate integral Lipschitz filters are stable to small perturbations in the underlying graphs as well as small perturbations in the time domain caused by time warping. Our analysis shows that small variations in the network topology and time evolution of a system does not significantly affect the performance of ST-GNNs. Numerical experiments with decentralized control systems showcase the effectiveness and stability of the proposed ST-GNNs.

## 1 INTRODUCTION

Graph Neural Networks (GNNs) are powerful convolutional architectures designed for network data. GNNs inherit all the favorable properties convolutional neural networks (CNNs) admit, while they also exploit the graph structure. An important feature of GNNs, germane to their success, is that the number of learnable parameters is independent of the size of the underlying networks. GNNs have manifested remarkable performance in a plethora of applications, e.g., recommendation systems (Ying et al., 2018; Wu et al., 2020), drug discovery and biology (Gainza et al., 2020; Strokach et al., 2020; Wu et al., 2021; Jiang et al., 2021), resource allocation in autonomous systems (Lima et al., 2020; Cranmer et al., 2021), to name a few.

Recently, there has been an increased interest in time-varying network data, as they appear in various systems and carry valuable dynamical information. This interest is mostly prominent in applications as decentralized controllers (Tolstaya et al., 2020; Gama et al., 2020b; Yang and Matni, 2021; Gama and Sojoudi, 2021), traffic-flow forecasting (Yu et al., 2018; Li et al., 2018; Fang et al., 2021), and skeleton-based action detection (Yan et al., 2018; Cheng et al., 2020; Pan et al., 2021). The state-of-the-art (SOTA) usually deploys an additional architecture side by side with a GNN so the latter learns patterns from the graph domain while the former works on the time sequences. One choice could be a CNN as in (Li et al., 2020; Isufi and Mazzola, 2021; Wang et al., 2021) or a recurrent neural network (RNN) as in (Seo et al., 2018; Nicolicioiu et al., 2019; Ruiz et al., 2020). However, these joint architectures are performed in a centralized manner in the sense that the up-to-date data of all nodes are given at any given time. While this is well suited for, but not limited to, the case of social networks and recommendation systems, many physical-network applications are decentralized in nature and suffer from time delays in delivering messages.

In this paper, we close the gap by developing a *causal* space-time convolutional architecture that jointly processes the graph-time underlying structure. That is, the convolutional layers preserve time delays in message passing. Our work is motivated by the following question. *Is it possible to transfer learning between signals and datasets defined over different space-time underlying structures?* This is a well motivated question since in practice we execute these architectures on graphs that are different from the graphs used in training and signals are sampled at different sampling rates between training and execution. The answer to the above question was provided for the case of static graph signals in

(Gama et al., 2020a), where the stability of traditional GNNs to graph perturbations was proved. In this work we give an affirmative answer to the above question in the case when time-varying graph signals are considered and space-time convolutional architectures are employed.

The contribution of this paper is twofold. First we introduce a novel convolutional architecture for time-varying graph signals, and second we prove its stability. Specifically, we provide a general definition of convolutions for any arbitrary shift operator and define a space-time shift operator (STSO) as the linear composition of the graph shift operator (GSO) and time-shift operator (TSO). We then introduce *space-time graph neural networks* (ST-GNNs), a cascade of layers that consist of space-time graph filters followed by point-wise nonlinear activation functions. The proposed ST-GNN allows processing continuous-time graph signals, which is pivotal in the stability analysis. Furthermore, we study the effect of *relative* perturbations on ST-GNNs and prove that small variations in the graph and/or irregularities in the sampling process of time-varying graph signals do not essentially affect the performance of the proposed ST-GNN architecture. Our theoretical findings are also supported by thorough experimental analysis based on decentralized control applications.

The rest of this paper is structured as follows. The related work is summarized in Section 2. Sections 3 and 4 present our contributions listed above. Numerical experiments and conclusions are presented in Sections 5 and 6, respectively. The proofs and extended experiments are provided in the appendices.

Notation: Bold small and large symbols, i.e. $\mathbf{x}$ and $\mathbf{X}$, denote vectors and matrices, respectively. Calligraphic letters mainly represent time-varying graph signals unless otherwise is stated.

## 2 RELATED WORK

**GNNs for Time-varying Graph Signals.** One of the early architectures was (Yan et al., 2018), which introduced a convolutional filter that aggregates information only from 1-hop neighbors. Relying on product graphs, (Isufi and Mazzola, 2021) introduced a convolutional architecture, where each node has access to the present and past data of the other nodes in the graph. A similar convolutional layer was studied in (Pan et al., 2021; Loukas and Foucard, 2016), and while restricted to a domain that preserves time delays, (Pan et al., 2021) shows that it is not necessarily less expressive. Meanwhile, (Wang et al., 2021) performs GNNs over graph signals at each time instance separately before a temporal convolution is performed at the GNN outputs to capture the evolution of graph embeddings. Graph RNNs are another architecture developed to deal with time-varying graph signals. For example, (Pareja et al., 2020) uses an RNN to evolve the GNN parameters over time. Moreover, (Hajiramezanali et al., 2019) combined the GRNN with a variational autoencoder (VGAE) to improve the former's expressive power. However, all these architectures did not take into account the physical restrictions in the form of time delays, associated with decentralized applications.

Similar to our work, other architectures considered the diffusion equation to form message-passing layers, e.g., (Xhonneux et al., 2020; Poli et al., 2021; Fang et al., 2021). These architectures parameterize the dynamics of graph signals with GNNs. In simple words, they learn a parameterization that helps find the current state variables from the previous ones. The architecture in (Chamberlain et al., 2021) is another example of these architectures, but learns the graph weights instead of a limited number of filter coefficients (i.e., it resembles the parameterization of graph attention networks in (Veličković et al., 2018)).

**Stability.** Deformation stability of CNNs was studied in (Bruna and Mallat, 2013; Bietti and Mairal, 2019). The notion of stability was then introduced to graph scattering transforms in (Gama et al., 2019; Zou and Lerman, 2020). In a following work, Gama et al. (2020a) presented a study of GNN stability to graph absolute and relative perturbations. Graphon neural networks was also analyzed in terms of its stability in (Ruiz et al., 2021). Moreover, (Pan et al., 2021) proves the stability to absolute perturbations of space-time graph scattering transforms.

## 3 SPACE-TIME GRAPH NEURAL NETWORKS

In this section, we present the proposed ST-GNN architecture for time-varying graph signals. First, we provide a general definition of convolutions and then we develop the space-time graph filters, which are the cornerstone of the ST-GNN architecture.

Our analysis starts with the homogeneous diffusion equation, which is defined with respect to the Laplacian differential operator $\mathcal{L}$

$$\frac{\partial \mathcal{X}(t)}{\partial t} = -\mathcal{L}\mathcal{X}(t). \tag{1}$$

Equation (1) describes processes of signals that evolve across the diffusion dimension, $t$. The diffused signal, $\mathcal{X}(t)$, is modeled as an abstract vector in a vector space $\mathbb{V}$. The vector space is associated with an inner product $\langle ., . \rangle_{\mathbb{V}}$, and is closed under addition and scalar multiplication, e.g., n-dimensional vector spaces and function spaces. The solution of (1) is $\mathcal{X}(t) = e^{-t\mathcal{L}}\mathcal{X}(0) = e^{-t\mathcal{L}}\mathcal{X}_0 = e^{-t\mathcal{L}}\mathcal{X}$ (the subscript is omitted in what follows), and describes the signal after time $t$ from the initial state.

### 3.1 CONVOLUTION

With the diffusion equation in our hands, we can now define the convolution operation as the linear combination of the diffusion sequence with a filter $h(t)$.

**Definition 1** (Convolution Operator). *For a linear shift-invariant filter with coefficients $h(t), t \geq 0$, the convolution between the filter and an input signal $\mathcal{X} \in \mathbb{V}$, with respect to a linear differential operator $\mathcal{L} : \mathbb{V} \to \mathbb{V}$, is defined as*

$$h *_D \mathcal{X} = \int_0^\infty h(t)e^{-t\mathcal{L}}\mathcal{X}dt, \tag{2}$$

*where $*_D$ denotes the convolution operator applied on signals with underlying structure $D$.*

Definition 1 is a valid convolution operator. With abuse of notation, we refer to $\mathcal{L}$ as the shift operator but it should be understood that the shift operation is executed using $e^{-\mathcal{L}}$. Definition 1 establishes a generalized form of convolutions for a wide variety of shift operators and signals. Next we discuss two convolution types of practical interest that follow directly from (2), i.e., time convolutions and graph convolutions.

**Time Convolutions:** They involve continuous-time signals, denoted by $x(\tau)$, that are elements of square-integrable function spaces, i.e, $x(\tau) \in L^2(\mathbb{R})$. The underlying structure of $x(\tau)$ is the real line, since the time variable $\tau \in \mathbb{R}$. In order to generate the traditional definition of time convolution, the *time shift operator* (TSO) is chosen as the differential operator, $\mathcal{L}_\tau = \partial/\partial\tau$. One can show that $e^{-t\partial/\partial\tau}x(\tau) = x(\tau - t)$, using the Taylor series,

$$e^{-t\partial/\partial\tau}x(\tau) = \sum_{n=0}^\infty \frac{(-t)^n}{n!} \cdot \frac{\partial^n}{\partial\tau^n}x(\tau) = \sum_{n=0}^\infty \frac{(u-\tau)^n}{n!} \cdot \frac{\partial^n}{\partial\tau^n}x(\tau) = x(u),$$

with $u = \tau - t$. In other words, the TSO performs a translation operation. Then (2) reduces to

$$x(\tau) *_\mathbb{R} h(\tau) = \int_0^\infty h(t)e^{-t\partial/\partial\tau}x(\tau)dt = \int_0^\infty h(t)x(\tau - t)dt, \tag{3}$$

which is indeed the traditional definition of convolutions between time signals. It is also worth noting that $e^{j\omega\tau}$, $\forall\omega \in [0, \infty)$ are eigenfunctions of the TSO $\mathcal{L}_\tau = \partial/\partial\tau$ with associated eigenvalues $j\omega$. This observation is pivotal in the stability analysis of section 4.

**Graph Convolutions:** They involve graph signals, $\mathbf{x} \in L^2(\mathbb{R}^N)$, that are $N$-dimensional square-summable vectors containing information associated with $N$ different nodes. The underlying structure of $\mathbf{x}$ is represented by a graph $\mathcal{G} = (\mathcal{V}, \mathcal{E}, \mathcal{W})$, where $\mathcal{V}$ is the set of nodes with cardinality $|\mathcal{V}| = N$, $\mathcal{E} \subseteq \mathcal{V} \times \mathcal{V}$ is the set of edges, and $\mathcal{W} : \mathcal{E} \to \mathbb{R}$ is a map assigning weights to the edges (Shuman et al., 2012). The *graph shift operator* (GSO), $\mathbf{S} \in \mathbb{R}^{N \times N}$, is a matrix representation of the graph sparsity, e.g., graph adjacency or Laplacian (Ortega et al., 2018). Applying $\mathbf{S}$ to a signal $\mathbf{x}$ results in a signal $\mathbf{Sx}$, where $[\mathbf{Sx}]_i$ represents aggregated data from all nodes $j$ that satisfy that $(i, j) \in \mathcal{E}$ or $(j, i) \in \mathcal{E}$. We focus on undirected graphs, and therefore, $\mathbf{S}$ is a real symmetric and diagonalizable matrix with a set of $N$ eigenvectors $\{\mathbf{v}_i\}_{i=1}^N$, each associated with a real eigenvalue $\lambda_i$.

The GSO is a linear operator $\mathbf{S} : L^2(\mathbb{R}^N) \to L^2(\mathbb{R}^N)$ that deals with discrete signals. Therefore, it is more common to discretize the diffusion process over graphs at times $kT_s, k \in \mathbb{Z}^+$, where $T_s$ is the sampling period. The convolution is then defined as

$$\mathbf{x} *_\mathcal{G} \mathbf{h} = \sum_{k=0}^{K-1} h_k e^{-k\mathbf{S}}\mathbf{x}, \tag{4}$$

where $\{h_k\}_k$ is the filter coefficients and $K$ is the number of filter taps. The finite number of taps is a direct consequence of Cayley-Hamilton theorem, and $\mathbf{h}$ is a finite-impulse response (FIR) filter. Equation (4) generalizes graph convolutions in the literature, where convolutions are defined as polynomials in the shift operator, i.e., $\mathbf{x} *_{\mathcal{G}} \mathbf{h} = \sum_{k=0}^{K-1} h_k \mathbf{S}^k \mathbf{x}$, (Sandryhaila and Moura, 2013). This definition executes the shift operation using the matrix $\mathbf{S}$ directly compared to $e^{-\mathbf{S}}$ in (4).

## 3.2 SPACE-TIME CONVOLUTION AND GRAPH FILTERS

The generalized definition of convolution in 1, as well as the definitions of time and graph convolutions in (3), (4) respectively, set the ground for the space-time convolution. Space-time convolutions involve signals that are time-varying and are also supported on a graph. They are represented by $\mathcal{X} \in L^2(\mathbb{R}^N) \otimes L^2(\mathbb{R})$, where $\otimes$ is the tensor product [see (Kadison and Ringrose, 1983; Grillet, 2007)] between the vector spaces of graph signals and continuous-time signals. In the following, calligraphic symbols (except $\mathcal{L}$) refer to time-varying graph signals unless otherwise is stated.

The *space-time shift operator* (STSO) is the linear operator that jointly shifts the signals over the space-time underlying support. Therefore, we choose the STSO to be the linear composition of the GSO and TSO, i.e., $\mathcal{L} = \mathbf{S} \circ \mathcal{L}_\tau$.

The convolution operator in Definition 1 can be specified with respect to the STSO for time-varying graph signals as

$$h *_{\mathcal{G} \times \mathbb{R}} \mathcal{X} = \int_0^\infty h(t) e^{-t\mathbf{S} \circ \mathcal{L}_\tau} \mathcal{X} dt =: \mathbf{H}(\mathbf{S}, \mathcal{L}_\tau) \mathcal{X}, \tag{5}$$

where $\mathbf{H}(\mathbf{S}, \mathcal{L}_\tau) = \int_0^\infty h(t) e^{-t\mathbf{S} \circ \mathcal{L}_\tau} dt$ represents the *space-time graph filter*. The frequency response of the filter is then derived as [see Appendix A]

$$\tilde{h}(\lambda, j\omega) = \int_0^\infty h(t) e^{-t(\lambda + j\omega)} dt. \tag{6}$$

**Remark 1.** *The space-time graph filters can be implemented distributively. For each node, the operator $\left(e^{-\mathbf{S} \circ \mathcal{L}_\tau}\right)^t$ aggregates data from the $t$-hop neighbors after they are shifted in time. This process can be done locally after each node acquires information from their neighbors. The time shift $t$ represents the time required for the data to arrive from the $t$-hop neighbors.*

## 3.3 SPACE-TIME GRAPH NEURAL NETWORKS

The natural extension of our previous analysis is to define the convolutional layer of ST-GNNs based on the space-time graph filters in (5). However, learning infinite-impulse response (IIR) filters is impractical and thus we employ FIR space-time graph filters, defined as $\mathbf{H}_d(\mathbf{S}, \mathcal{L}_\tau) = \sum_{k=0}^{K-1} h_k e^{-kT_s \mathbf{S} \circ \mathcal{L}_\tau}$. The ST-GNN architecture employs a cascade of $L$ layers, each of which consists of a bank of space-time graph filters followed by a pointwise nonlinear activation functions, denoted by $\sigma$. The input to layer $\ell$ is the output of the previous layer, $\mathcal{X}_{\ell-1}$, and the output of the $\ell$th layer is written as

$$\mathcal{X}_\ell^f = \sigma \left( \sum_{g=1}^{F_{\ell-1}} \sum_{k=0}^{K-1} h_{k\ell}^{fg} e^{-kT_s \mathbf{S} \circ \mathcal{L}_\tau} \mathcal{X}_{\ell-1}^g \right), \tag{7}$$

for each feature $\mathcal{X}_\ell^f$, $f = 1, \ldots, F_\ell$. The number of features at the output of each layer is denoted by $F_\ell$, $\ell = 1, \ldots, L$. A concise representation of the ST-GNN can be written as $\Phi(\mathcal{X}; \mathcal{H}, \mathbf{S} \circ \mathcal{L}_\tau)$, where $\mathcal{H}$ is the set of all learnable parameters $\{h_{k\ell}^{fg}\}_{k,\ell}^{f,g}$. An interesting remark is that the time-varying graph signals processed in (7) are continuous in time, but learning a finite number of parameters.

**Remark 2.** *The ST-GNNs are causal architectures, that is, the architecture does not allow the nodes to process data that are still unavailable to them. This is achieved by delaying the data coming from the $k$-hop neighbors by $k$ time steps [c.f. (7)]. Therefore, each node has access to its own up-to-date data and outdated data from their neighbors.*

## 4 STABILITY TO PERTURBATIONS

In this section, we perform a stability analysis of our proposed ST-GNN architecture. In particular, we characterize the difference between the filters $\mathbf{H}(\mathbf{S}, \mathcal{L}_\tau)$ and $\mathbf{H}(\hat{\mathbf{S}}, \hat{\mathcal{L}}_\tau)$, where $\hat{\mathbf{S}}$ and $\hat{\mathcal{L}}_\tau$ are perturbed versions of the graph and time shift operators, respectively.

## 4.1 Perturbation Models

Before we study the effect of graph and time perturbations on the stability of our proposed ST-GNN, we first need to introduce a perturbation model in space and time. For graph perturbations, we follow the relative perturbation model proposed in (Gama et al., 2020a):

$$\mathbf{P}_0^T \hat{\mathbf{S}} \mathbf{P}_0 = \mathbf{S} + \mathbf{S}\mathbf{E} + \mathbf{E}\mathbf{S}, \tag{8}$$

where $\hat{\mathbf{S}}$ is the perturbed GSO, $\mathbf{E}$ is the error matrix and $\mathbf{P}_0$ is a permutation matrix. Taking a closer look in equation (8) we observe that the entries of $\mathbf{S}$ and $\hat{\mathbf{S}}$ become more dissimilar as the norm $\|\mathbf{E}\|$ increases. The dissimilarity is measured by the difference $[\mathbf{S}]_{ij} - [\mathbf{P}_0^T \hat{\mathbf{S}} \mathbf{P}_0]_{ij} = [\mathbf{S}\mathbf{E}]_{ij} + [\mathbf{E}\mathbf{S}]_{ij}$, where each summand is a weighted sum of the edges that are connected to nodes $i$ and $j$, respectively. For nodes $i$ and $j$ with high node degrees, the sum includes a high number of non-zero entries from $\mathbf{S}$, which leads to perturbations of high volume. Consequently, the considered graph perturbations are *relative* to node degrees. It is also worth noticing that equation (8) is invariant to node permutations of the GSO. This is due to the use of $\mathbf{P}_0$ which reconciles for node relabelling of perturbed graphs (see Section 4.2).

While there has been plenty of work regarding graph perturbation models, time perturbation in the context of graph neural networks is not well studied. In this paper we fill this gap and propose a practical model for time perturbations. In particular, we study irregularities that appear in the process of sampling continuous-time signals, $x(\tau)$. It is often the case that sampling is imperfect and not exactly equispaced, i.e., the discrete signals are captured at times $kT_s \pm z(kT_s)$, where $k \in \mathbb{Z}, T_s \in \mathbb{R}$ is the sampling period and $z : \mathbb{R} \to \mathbb{R}$ is a differentiable function. To model this phenomenon, we use time-warping operations. In particular, we consider a perturbed timeline $\hat{\tau} = \tau + z(\tau)$, and we observe the signal $x(\hat{\tau}) = x(\tau + z(\tau))$ instead of $x(\tau)$. The diffusion process in (1) for a continuous-time signal, over the perturbed timeline, can then be cast as

$$\frac{\partial}{\partial t} x(t, \hat{\tau}) = -\frac{\partial}{\partial \hat{\tau}} x(t, \hat{\tau}) \Rightarrow \frac{\partial}{\partial t} x(t, \tau + z(\tau)) = -(1 + z'(\tau)) \frac{\partial}{\partial \tau} x(t, \tau + z(\tau)). \tag{9}$$

Now, let $\xi(\tau) = z'(\tau)$, $\mathcal{L}_\tau = \frac{\partial}{\partial \tau}$, and $\hat{\mathcal{L}}_\tau = \frac{\partial}{\partial \hat{\tau}}$. Then the perturbed TSO takes the form

$$\hat{\mathcal{L}}_\tau = (1 + \xi(\tau)) \ \mathcal{L}_\tau. \tag{10}$$

The effect of time perturbation is controlled by $\xi(\tau)$. As the norm $\|\xi(\tau)\|_2$ grows, the dissimilarity between the original and perturbed time shift operators increases. This shows that the considered time perturbation model is relative to the rate of change of $z(\tau)$; faster changes in $z(\tau)$ produce more significant perturbations (i.e., longer time shifts), whereas smoother choices of $z(\tau)$ yield lower values of $\|\xi(\tau)\|_2$ and lower perturbation levels. The time convolution in (3) with a perturbed TSO becomes

$$x(\tau + z(\tau)) *_\mathbb{R} h(\tau) = \int_0^\infty h(t) e^{-t(1 + \xi(\tau))\mathcal{L}_\tau} x(\tau + z(\tau)) dt$$

$$= \int_0^\infty h(t) e^{-t(1 + \xi(\tau))\mathcal{L}_\tau} e^{z(\tau)\mathcal{L}_\tau} x(\tau) dt, \tag{11}$$

since $x(\tau + z(\tau)) = e^{z(\tau)\mathcal{L}_\tau} x(\tau)$, which models a signal translation $e^{z(\tau)\mathcal{L}_\tau}$ caused by time perturbations. Moreover, the perturbed TSO $e^{-t(1 + \xi(\tau))\mathcal{L}_\tau}$ models a time-varying shift of $t(1 + \xi(\tau))$ when $\xi(\tau)$ is not a constant (which is the case in our analysis).

## 4.2 Joint Operator-Distance Modulo

In order to proceed with the stability analysis, we need to define the operators used to measure the effect of perturbations. We start with the measure of graph perturbations as we utilize the *operator distance modulo permutation*, $\|.\|_\mathcal{P}$, introduced in (Gama et al., 2020a). For any two graph operators $\mathbf{A}$ and $\hat{\mathbf{A}} : L^2(\mathbb{R}^N) \to L^2(\mathbb{R}^N)$, the distance between them can be calculated as

$$\|\mathbf{A} - \hat{\mathbf{A}}\|_\mathcal{P} = \min_{\mathbf{P} \in \mathcal{P}} \max_{\mathbf{x}: \|\mathbf{x}\|_2 = 1} \|\mathbf{A}\mathbf{x} - \mathbf{P}^T \hat{\mathbf{A}} \mathbf{P} \mathbf{x}\|_2, \tag{12}$$

where $\mathcal{P}$ is the set of all permutation matrices. This distance measures how close an operator $\hat{\mathbf{A}}$ to being a permuted version of $\mathbf{A}$. The distance in (12) is minimum (i.e., 0) when the two graphs are permuted versions of each other.

Next, we define a new *operator distance modulo translation* for time operations:

**Definition 2** (Operator Distance Modulo Translation). *Given the linear time operators $\mathcal{B}$ and $\hat{\mathcal{B}} : L^2(\mathbb{R}) \to L^2(\mathbb{R})$, define the operator distance modulo translation as*

$$\|\mathcal{B} - \hat{\mathcal{B}}\|_{\mathcal{T}} = \min_{s \in \mathbb{R}} \max_{x : \|x\|_2 = 1} \left\| \mathcal{B}x(\tau) - (e^{-s\mathcal{L}_\tau} \circ \hat{\mathcal{B}})x(\tau) \right\|_2. \tag{13}$$

The expression in (13) measures how far or close two time operators are, to being translated versions of each other. In the case where $\hat{\mathcal{L}}_\tau = (1 + \xi(\tau))\ \mathcal{L}_\tau$ as in (10), the distance between $\mathcal{B} = e^{-t\mathcal{L}_\tau}$ and $\hat{\mathcal{B}} = e^{-t\hat{\mathcal{L}}_\tau}$ reduces to

$$\|e^{-t\mathcal{L}_\tau} - e^{-t\hat{\mathcal{L}}_\tau}\|_{\mathcal{T}} = \min_{s \in \mathbb{R}} \max_{x : \|x\|_2 = 1} \|x(\tau - t) - x(\tau - s - t - t\xi(\tau - s))\|_2. \tag{14}$$

Since we are dealing with time-varying graph signals and joint time and graph perturbations, we combine the two previously defined distances and introduce the *operator distance modulo permutation-translation*.

**Definition 3** (Operator Distance Modulo Permutation-Translation). *Given the graph and time operators $\mathcal{A}, \hat{\mathcal{A}} : L^2(\mathbb{R}^N) \to L^2(\mathbb{R}^N)$, $\mathcal{B}, \hat{\mathcal{B}} : L^2(\mathbb{R}) \to L^2(\mathbb{R})$, define the operator distance modulo permutation-translation as*

$$\|\mathbf{A} \circ \mathcal{B} - \hat{\mathbf{A}} \circ \hat{\mathcal{B}}\|_{\mathcal{P},\mathcal{T}} = \min_{\mathbf{P} \in \mathcal{P}} \min_{s \in \mathbb{R}} \max_{\mathcal{X} : \|\mathcal{X}\|_F = 1} \left\| (\mathbf{A} \circ \mathcal{B})\mathcal{X} - (\mathbf{P}^T \hat{\mathbf{A}} \mathbf{P}) \circ (e^{-s\mathcal{L}_\tau} \circ \hat{\mathcal{B}})\mathcal{X} \right\|_F. \tag{15}$$

### 4.3 STABILITY PROPERTIES OF SPACE-TIME GRAPH FILTERS

The last step before we present our stability results is to discuss integral Lipschitz filters, which are defined as:

**Definition 4** (Multivariate Integral Lipschitz Filters). *Let the frequency response of a multivariate filter be $\tilde{h}(\mathbf{w}) : \mathbb{R}^n \to \mathbb{R}$. The filter is said to be integral Lipschitz if there exists a constant $C > 0$ such that for all $\mathbf{w}_1$ and $\mathbf{w}_2$,*

$$\left| \tilde{h}(\mathbf{w}_2) - \tilde{h}(\mathbf{w}_1) \right| \leq 2C \frac{\|\mathbf{w}_2 - \mathbf{w}_1\|_2}{\|\mathbf{w}_2 + \mathbf{w}_1\|_2}. \tag{16}$$

The condition in (16) requires the frequency response of the filter to be Lipschitz over line segments defined by arbitrary $\mathbf{w}_1$ and $\mathbf{w}_2 \in \mathbb{R}^n$, with a Lipschitz constant that is inversely proportional to their midpoint. For differentiable filters the above condition reduces to $\left| \frac{\partial}{\partial \zeta} \tilde{h}(\mathbf{w}_1) \right| \leq C \frac{1}{\|\mathbf{w}_1\|_2}$ for every $\zeta$ that is a component of $\mathbf{w}_1$. The proposed space-time graph filters admit a frequency response which is bivariate with variables $\lambda$ and $j\omega$ [cf. (6)]. Following Definition 4, a space-time graph filter $\tilde{h}(\lambda, j\omega)$ is integral Lipschitz, if

$$|\lambda + j\omega| \cdot \left| \frac{\partial}{\partial \zeta} \tilde{h}(\lambda, j\omega) \right| \leq C, \ \forall \zeta \in \{\lambda, j\omega\}. \tag{17}$$

Note that the vector $\mathbf{w}_1$ in this case is given as $[\lambda, \omega]^T$ with $\|\mathbf{w}_1\|_2 = |\lambda + j\omega|$. The conditions in (17) indicate that the frequency response of an integral Lipschitz space-time graph filter can vary rapidly at low frequencies close to 0. Therefore, the filter can *discriminate* between close low-frequency spectral components, but is more flat at high frequencies, prohibiting discriminability between the spectral features in these bands. Note that when we mention low frequencies in the context of space time graph filters (or ST-GNNs later), we refer to low values of $\lambda$ and $\omega$, whereas high frequencies correspond to high values of $\lambda$ or $\omega$.

**Proposition 1** (Stability to Graph Perturbations). *Consider a space-time graph filter $\mathbf{H}$ along with graph shift operators $\mathbf{S}$ and $\hat{\mathbf{S}}$, and a time shift operator $\mathcal{L}_\tau$. If it holds that*

*(A1) the GSOs are related by $\mathbf{P}_0^T \hat{\mathbf{S}} \mathbf{P}_0 = \mathbf{S} + \mathbf{S}\mathbf{E} + \mathbf{E}\mathbf{S}$ with $\mathbf{P}_0$ being a permutation matrix,*
*(A2) the error matrix $\mathbf{E}$ has a norm $\|\mathbf{E}\| \leq \epsilon_s$, and an eigenvector misalignment $\delta$ relative to $\mathbf{S}$, and*
*(A3) the filter $\mathbf{H}(\mathbf{S}, \mathcal{L}_\tau)$ is an integral Lipschitz filter with a Lipschitz constant $C > 0$, then*

*the difference between the space-time graph filters $\mathbf{H}(\mathbf{S}, \mathcal{L}_\tau)$ and $\mathbf{H}(\hat{\mathbf{S}}, \mathcal{L}_\tau)$ is characterized by*

$$\|\mathbf{H}(\mathbf{S}, \mathcal{L}_\tau) - \mathbf{H}(\hat{\mathbf{S}}, \mathcal{L}_\tau)\|_{\mathcal{P}} \leq 2C\epsilon_s \left( 1 + \delta\sqrt{N} \right) + \mathcal{O}(\epsilon_s^2), \tag{18}$$

*where $\delta = (\|\mathbf{U} - \mathbf{V}\| + 1)^2 - 1$, and $N$ is the number of nodes in $\mathbf{S}$.*

Proposition 1 states that space-time graph filters are stable to graph perturbations of order $\mathcal{O}(\epsilon_s)$ with a stability constant $2C(1 + \delta\sqrt{N})$. The constant is uniform for all graphs of the same size and depends on the Lipschitz constant of the filter. The bound is proportional to the eigenvector misalignment between the GSO and the error matrix. The misalignment is bounded and does not grow with the graph size as $\delta \le (\|\mathbf{U}\| + \|\mathbf{V}\| + 1)^2 - 1 = 8$ with $\mathbf{U}$ and $\mathbf{V}$ being unitary matrices. Now, consider a type of perturbations known as graph dilation, where the edge weights stretch by a value $\epsilon_s$, i.e., the GSO becomes $\hat{\mathbf{S}} = (1 + \epsilon_s)\mathbf{S}$. In this case, the eigenvalues of the GSO also stretch by $\epsilon_s$, but the eigenvectors are no longer misaligned, i.e., $\delta = 0$. The bound for graph dilation then reduces to $2C\epsilon$ and does not depend on the structure of the underlying graph. Therefore, it is inferred that the bound in (18) is split into (i) a term that reflects the difference in eigenvalues between $\mathbf{S}$ and $\hat{\mathbf{S}}$, and (ii) a term that arises from the eigenvector misalignment. Note that the stability constant is not affected by any property of the TSO, and consequently has the same form of the one derived by Gama et al. (2020a) for traditional graph filters.

**Proposition 2** (Stability to Time Perturbations). *Consider a space-time graph filter $\mathbf{H}$ along with time shift operators $\mathcal{L}_\tau$ and $\hat{\mathcal{L}}_\tau$, and a graph shift operator $\mathbf{S}$. If it holds that*

*(A1) the TSOs are related by $\hat{\mathcal{L}}_\tau = (1 + \xi(\tau))\,\mathcal{L}_\tau$,*
*(A2) the error function $\xi(\tau)$ is infinitely differentiable with a norm $\|\xi(\tau)\|_2 \le \kappa\epsilon_\tau$, and the norm of the $m$th-order derivative$\|\xi^{(m)}(\tau)\|_2$ is of order $\mathcal{O}(\epsilon_\tau^{m+1})$ where $\kappa$ is an absolute constant, and*
*(A3) the filter $\mathbf{H}(\mathbf{S}, \mathcal{L}_\tau)$ is an integral Lipschitz filter with a Lipschitz constant $C > 0$, then*

*the difference between the space-time graph filters $\mathbf{H}(\mathbf{S}, \mathcal{L}_\tau)$ and $\mathbf{H}(\mathbf{S}, \hat{\mathcal{L}}_\tau)$ satisfies*

$$\|\mathbf{H}(\mathbf{S}, \mathcal{L}_\tau) - \mathbf{H}(\mathbf{S}, \hat{\mathcal{L}}_\tau)\|_\mathcal{T} \le C\kappa\epsilon_\tau + \mathcal{O}(\epsilon_\tau^2). \tag{19}$$

Similarly to Proposition 1, the filter difference is bounded by a constant proportional to the size of perturbation, i.e., $\|\xi(\tau)\|_2$, and the bound is also affected by the Lipschitz constant of the filter. The perturbed TSO shares the same eigenfunctions with the original TSO since $\hat{\mathcal{L}}_\tau e^{j\omega\tau} = j\omega(1 + \xi(\tau))e^{j\omega\tau}$. Therefore, the difference between the filters only arises from the difference in eigenvalues. It is worth mentioning that the assumption $(A2)$ of Proposition 2 is rather reasonable. One example of a time-warping function $z(\tau)$ that satisfies this assumption is $z(\tau) = \sqrt{\epsilon_\tau}\cos(\epsilon_\tau\tau)e^{-\epsilon_\tau\tau}$. The error function is then $\xi(\tau) = z'(\tau) = -\epsilon_\tau^{3/2}(\sin\epsilon_\tau\tau + \cos\epsilon_\tau\tau)e^{-\epsilon_\tau\tau}$ with a norm of $\sqrt{3/4\epsilon_\tau}$. The first derivative of the error function is $\xi'(\tau) = -2\epsilon_\tau^{5/2}\cos(\epsilon_\tau\tau)e^{-\epsilon_\tau\tau}$ and its norm is of order $\mathcal{O}(\epsilon_\tau^2)$, which increases exponentially with the order of the derivatives.

In Theorem 1, we consider joint time and graph perturbations, and characterize the difference between the filters under the STSO, $\mathbf{S} \circ \mathcal{L}_\tau$, and its perturbed version $\hat{\mathbf{S}} \circ \hat{\mathcal{L}}_\tau$.

**Theorem 1** (Space-Time Graph Filter Stability). *Under the assumptions of Propositions 1 and 2, the distance between the space-time graph filters $\mathbf{H}(\mathbf{S}, \mathcal{L}_\tau)$ and $\mathbf{H}(\hat{\mathbf{S}}, \hat{\mathcal{L}}_\tau)$ satisfies*

$$\|\mathbf{H}(\mathbf{S}, \mathcal{L}_\tau) - \mathbf{H}(\hat{\mathbf{S}}, \hat{\mathcal{L}}_\tau)\|_{\mathcal{P}, \mathcal{T}} \le 2C\epsilon_s\left(1 + \delta\sqrt{N}\right) + C\kappa\epsilon_\tau + \mathcal{O}(\epsilon^2), \tag{20}$$

*where $\epsilon = \max\{\epsilon_s, \epsilon_\tau\}$.*

In order to keep this stability bound small, space-time graph filters should be designed with small Lipschitz constant $C$. This comes at the cost of filter discriminability at high frequencies [cf. (17)], whereas at low frequencies, the integral Lipschitz filters are allowed to vary rapidly. Thus Theorem 1 shows that space-time graph filters are stable and discriminative at low frequencies but cannot be both stable and discriminative at higher frequencies.

## 4.4 Stability Properties of Space-Time Graph Neural Networks

Theorem 2 establishes the stability of the proposed ST-GNN architecture.

**Theorem 2** (ST-GNNs Stability). *Consider an $L$-layer ST-GNN $\mathbf{\Phi}(\cdot; \mathcal{H}, \mathbf{S} \circ \mathcal{L}_\tau)$ with a single feature per each layer. Consider also the GSOs $\mathbf{S}$ and $\hat{\mathbf{S}}$, and the TSOs $\mathcal{L}_\tau$ and $\hat{\mathcal{L}}_\tau$. If*

*(A1) the filters at each layer are integral Lipschitz with a Lipschitz constant $C > 0$ and have unit operator norms, i.e., $\|\mathbf{H}_\ell(\mathbf{S}, \mathcal{L}_\tau)\| = 1, \forall \ell = 1, \ldots, L$,*
*(A2) the nonlinearirties $\sigma$ are Lipschitz-continuous with a Lipschitz constant of 1, i.e., $\|\sigma(\mathbf{x}_2) - \sigma(\mathbf{x}_1)\|_2 \le \|\mathbf{x}_2 - \mathbf{x}_1\|_2$,*

*(A3) the GSOs satisfy (A1) and (A2) of Proposition 1, and the TSOs satisfy (A1) and (A2) of Proposition 2, then*

$$\|\mathbf{\Phi}(\cdot; \mathcal{H}, \mathbf{S} \circ \mathcal{L}_\tau) - \mathbf{\Phi}(\cdot; \mathcal{H}, \hat{\mathbf{S}} \circ \hat{\mathcal{L}}_\tau)\|_{\mathcal{P},\mathcal{T}} \leq 2CL\epsilon_s \left(1 + \delta\sqrt{N}\right) + CL\kappa\epsilon_\tau + \mathcal{O}(\epsilon^2), \quad (21)$$

*where $\epsilon = \max\{\epsilon_s, \epsilon_\tau\}$.*

Theorem 2 shows that the stability bound of ST-GNNs depends on the number of layers $L$, in addition to the factors affecting the stability of space-time graph filters. Compared to space-time graph filters, ST-GNNs can be both stable and discriminative. This is due to the nonlinearities applied at the end of each layer. The effect of the pointwise nonlinearity is that it demodulates (i.e., spills) the energy of the higher-frequency components into lower frequencies [see (Gama et al., 2020a)]. Once the frequency content gets spread over the whole spectrum by the consecutive application of the nonlinearities, the integral Lipschiz filters in the deeper layers become discriminative. The stability/discriminability property of ST-GNNs is a pivotal reason why they outperform space-time graph filters.

## 5  NUMERICAL EXPERIMENTS

We examine two decentralized-controller applications: flocking, and motion planning. Specifically, we are given a network of $N$ agents, where agent $i$ has position $\mathbf{p}_{i,n} \in \mathbb{R}^2$, velocity $\mathbf{v}_{i,n} \in \mathbb{R}^2$ and acceleration $\mathbf{u}_{i,n} \in \mathbb{R}^2$ at time steps $n < T$. The agents collaborate to learn controller actions and complete a specific task. For each task, the goal is to learn a controller $\mathbf{U}$ that imitates an optimal centralized controller $\mathbf{U}^*$. Therefore, we parameterize $\mathbf{U}$ with ST-GNNs and aim to find $\mathcal{H}^*$ that satisfies

$$\mathcal{H}^* = \arg\min_{\mathcal{H} \in \mathbb{H}} \frac{1}{M} \sum_{m=0}^{M} \ell\Big(\mathbf{\Phi}(\mathbf{X}_m; \mathcal{H}, \mathcal{L}_m) - \mathbf{U}_m^*\Big), \quad (22)$$

where $\mathbb{H}$ is the set of all possible parameterizations, $\ell(.)$ is the mean-squared loss function, $\mathcal{L}_m = (\mathbf{S} \circ \mathcal{L}_\tau)_m$ is the STSO of the m-*th* example, and $M$ is the size of the training dataset. The input signal $\mathbf{X}_m \in \mathbb{R}^{q \times N \times T}$ contains $q$ state variables of $N$ agents, e.g. the position and velocity, over $T$ time steps. Detailed description of the ST-GNN implementation can be found in Appendix G.

**A. Network Consensus and Flocking.** A swarm of $N$ agents collaborates to learn a reference velocity $\mathbf{r}_n \in \mathbb{R}^2, \forall n$, according to which the agents move to avoid collisions. Ideally the reference velocity and the agent velocity $\mathbf{v}_{i,n}$, should be the same, which is not the case in practice since each agent observes a noisy version of the reference velocity $\tilde{\mathbf{r}}_{i,n}$. Therefore the state variables (inputs) are the estimated velocities $\{\mathbf{v}_{i,n}\}_{i,n}$, the observed velocities $\{\tilde{\mathbf{r}}_{i,n}\}_{i,n}$ and the relative masses $\{\mathbf{q}_{i,n} | \mathbf{q}_{i,n} = \sum_{j \in \mathcal{N}_i}(\mathbf{p}_{i,n} - \mathbf{p}_{j,n})\}_{i,n}$, where $\mathcal{N}_i$ is the set of direct neighbors of the $i$-th agent. We train an ST-GNN to predict agent accelerations $\{\mathbf{u}_{i,n}\}_{i,n}$ according to which the agents will move to a new position with certain velocity. Further details can be found in Appendix G.

**Experiment 1.** First we organize 100 agents in a mesh grid, but the agents are not allowed to move. The reason is that we want to test the ST-GNN on a setting where the graph is not changing over time. Note that although the agents do not move they still estimate an acceleration and therefore a velocity according to which they should move. In simple words the goal is to estimate the optimal acceleration of each agent in the case they would be required to move. Consequently, the training and testing is executed over the same graph but with different state variables and outputs. Fig. 1 (Left) illustrates the mean of agent velocities, defined as $\mathbf{v}_n = \sum_{i=1}^{N} \mathbf{v}_{i,n}$, along with the reference velocity $\mathbf{r}_n$. We observe that the agents succeed to reach consensus and follow the reference velocity. We also execute the previously trained ST-GNN on a setting with perturbed graph and time shift operators according to (8) and (10), respectively. Specifically, the error matrix $\mathbf{E}$ is diagonal with $\|\mathbf{E}\| \leq \epsilon$, and $z(\tau) = \sqrt{\epsilon}\cos(\epsilon\tau)e^{-\epsilon\tau}$. Fig. 1 (Middle) shows that the relative RMSE of the outputs $\{\mathbf{u}_{i,n}\}_{i,n}$, defined as $\|\mathbf{\Phi}(\mathbf{X}; \mathcal{H}, \mathbf{S} \circ \mathcal{L}_\tau) - \mathbf{\Phi}(\mathbf{X}; \mathcal{H}, \hat{\mathbf{S}} \circ \hat{\mathcal{L}}_\tau)\|_F / \|\mathbf{\Phi}(\mathbf{X}; \mathcal{H}, \mathbf{S} \circ \mathcal{L}_\tau)\|_F$. We observe that the relative RMSE is linearly proportional to the perturbation size $\epsilon$, following the results of Theorem 2.

**Experiment 2.** We now allow 50 agents to move and form a communication network that changes with their movement, i.e., the graph of the agents varies over time. Consequently, the ST-GNN is trained on a dynamic graph [see Appendix G]. Fig. 1 (Right) shows the difference between the agent and reference velocities, $\mathbf{v}_{i,n}$ and $\mathbf{r}_n$, averaged over $N$ agents. The small standard deviation in the velocity difference (represented in red) indicates that the agents reached a consensus on their velocities. The figure also shows that the ST-GNN outperforms the decentralized policy, described in Appendix G. This experiment demonstrates the effectiveness of the proposed ST-GNN architecture

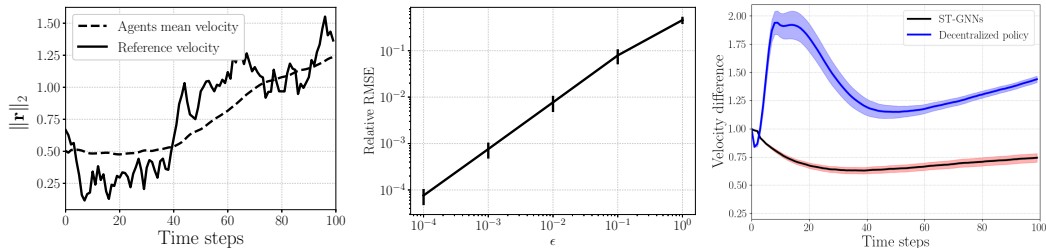

Figure 1: (Left) The mean of the velocity estimates by the agents compared to the reference velocity in a test example. (Middle) The relative RMSE in the ST-GNN outputs under perturbations to the underlying space-time topology. (Right) The mean difference between agent and reference velocities.

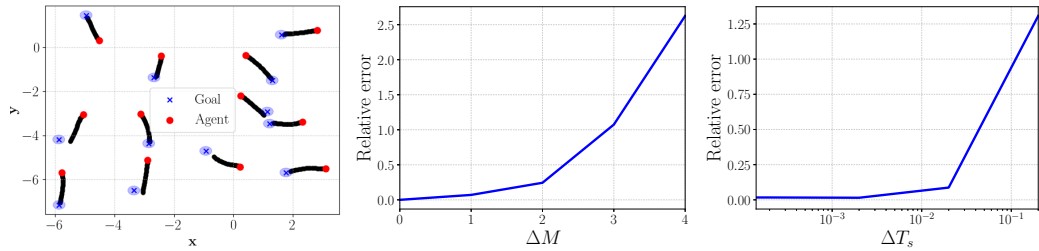

Figure 2: (Left) Three-second agent trajectories of a test example. (Middle) Relative error caused by using different network densities. (Right) Relative error caused by using different sampling time $T_s$.

on dynamic graphs, even though our theoretical analysis is derived for fixed graphs. Further results are presented in Appendix G.

**B. Unlabeled Motion Planning.** The goal of this task is to assign $N$ unlabeled agents to $N$ target locations. The agents collaborate to find their free-collision trajectories to the assigned targets. We train an ST-GNN to learn the control accelerations $\{\mathbf{u}_{i,n}\}_{i,n}$ according to (22). The state variables for agent $i$ are position $\{\mathbf{p}_{i,n}\}_n$, velocity $\{\mathbf{v}_{i,n}\}_n$, the position and velocity of agent's $M$-nearest neighbors, and the position of the $M$-nearest targets at each time step $n$. The underlying graph is constructed as $(i,j) \in \mathcal{E}_n$ if and only if $j \in \mathcal{N}_{i,n}$ or $i \in \mathcal{N}_{j,n}$, where $\mathcal{N}_{i,n}$ is the set of the $M$-nearest neighbors of agent $i$ at time step $n$. The rest of the training parameters are shown in Appendix G.

We executed the learned ST-GNN on a test dataset that is designed according to the parameter $M$ and sampling time $T_s$. A snapshot of the predicted trajectories are shown in Fig. 2 (Left). We observe that the agents approach their target goals without collisions. The average distance between the agent's final position and the desired target is, $\hat{d}_{pg} = 0.524$ with variance equal to $0.367$. We now test the sensitivity of ST-GNNs to the right choice of $M$ and $T_s$ in the testing phase. Fig. 2 (Middle) shows the relative error, which is calculated as $(\hat{d}_{pg,\text{pert}} - \hat{d}_{pg,\text{org}})/\hat{d}_{pg,\text{org}}$, when we change $M$. The error increases with $\Delta M$ because changing the neighborhood size induces a change in the underlying graph. According to our theoretical analysis, the error increases with the size of graph perturbations, which matches the results in Fig. 2 (Middle). The same remark can be observed in Fig. 2 (Right) when we change the sampling time, which induces a change in the underlying time structure.

## 6 CONCLUSIONS

In this paper we developed a novel ST-GNN architecture, tailored for time-varying signals that are also supported on a graph. We showed that the proposed architecture is stable under both graph and time perturbations under certain conditions. Our conditions are practical and provide guidelines on how to design space-time graph filters to achieve desirable stability. Our theoretical analysis is supported by strong experimental results. Simulations on the tasks of decentralized flocking and unlabeled motion planning corroborated our theoretical results and demonstrated the effectiveness of the proposed ST-GNN architecture.

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

## A  (PROOF OF) FREQUENCY RESPONSE OF SPACE-TIME GRAPH FILTERS

We start from the abstract representation of signal $\mathcal{X} \in \mathbb{V}$, on which we apply a shift operator $\mathcal{L}: \mathbb{V} \to \mathbb{V}$. Then we can write the signal $\mathcal{X}$ as

$$\mathcal{X} = \int_i \langle \mathcal{X}, \mathcal{U}(i) \rangle_{\mathbb{V}} \, \mathcal{U}(i) di, \tag{23}$$

where $\{\mathcal{U}(i)\}_i$ is the set of the orthonormal eigenfunctions of $\mathcal{L}$ that spans the space $\mathbb{V}$. The form in (23) is a reminiscence of the inverse Fourier transform and the scalar values $\langle \mathcal{X}, \mathcal{U}(i) \rangle_{\mathbb{V}}$ are the Fourier coefficients. In classical signal processing, the Fourier coefficients represent the spectral components of the signal $\mathcal{X}$.

Now, we consider the example of time-varying graph signals, $\mathcal{X} \in L^2(\mathbb{R}^N) \otimes L^2(\mathbb{R})$, which are constructed as the tensor product of a continuous-time signal and a graph signal. Shifting the signal $\mathcal{X}$ is executed using the STSO, $\mathbf{S} \circ \mathcal{L}_\tau$, which is the linear composition of the GSO and TSO. The eigenfunctions of the STSO can then be defined as the tensor product $\mathbf{v}_i \otimes e^{j\omega\tau}$, where $\mathbf{v}_i$ is an eigenvector of the GSO and $e^{j\omega\tau}$ is an eigenfunction of the TSO. Recall that the eigenfunction is itself a vector in the space $L^2(\mathbb{R}^N) \otimes L^2(\mathbb{R})$, and thus applying the STSO on $\mathbf{v}_i \otimes e^{j\omega\tau}$ means to jointly shift the the graph vector and the continuous-time function. In other words, we have

$$e^{-\mathbf{S} \circ \mathcal{L}_\tau}(\mathbf{v}_i \otimes e^{j\omega\tau}) = e^{-\mathbf{S}} \mathbf{v}_i \otimes e^{-\mathcal{L}_\tau} e^{j\omega\tau} = e^{-(\lambda_i + j\omega)}(\mathbf{v}_i \otimes e^{j\omega\tau}). \tag{24}$$

Thus we conclude that every eigenfunction $\mathbf{v}_i \otimes e^{j\omega\tau}$ of the STSO is associated with an eigenvalue $\lambda_i + j\omega$.

Defining the eigenfunctions of the STSO allows us to re-write (23) for time-varying graph signals as

$$\mathcal{X} = \frac{1}{2\pi} \sum_{i=1}^{N} \int_0^\infty \tilde{\mathcal{X}}(\lambda_i, j\omega)(\mathbf{v}_i \otimes e^{j\omega\tau}) d\omega, \tag{25}$$

where $\tilde{\mathcal{X}}(\lambda_i, j\omega)$ is the spectral component of the signal $\mathcal{X}$ at the frequency pair $(\lambda_i, \omega)$, and the constant $\frac{1}{2\pi}$ is to normalize the eigenfunctions. It is inferred that the frequency domain of time-varying graph signals is bivariate. Considering the space-time graph filter $\mathbf{H}(\mathbf{S}, \mathcal{L}_\tau)$, the filter output can be expressed as

$$\mathcal{Y} = \mathbf{H}(\mathbf{S}, \mathcal{L}_\tau)\mathcal{X} = \int_0^\infty h(t) e^{-t\mathbf{S} \circ \mathcal{L}_\tau} \mathcal{X} dt$$

$$\overset{(a)}{=} \frac{1}{2\pi} \sum_{i=1}^{N} \int_0^\infty \tilde{\mathcal{X}}(\lambda_i, j\omega) \int_0^\infty h(t) e^{-t\mathbf{S} \circ \mathcal{L}_\tau}(\mathbf{v}_i \otimes e^{j\omega\tau}) dt d\omega \tag{26}$$

$$\overset{(b)}{=} \frac{1}{2\pi} \sum_{i=1}^{N} \int_0^\infty \tilde{\mathcal{X}}(\lambda_i, j\omega) \int_0^\infty h(t) e^{-t(\lambda_i + j\omega)} dt (\mathbf{v}_i \otimes e^{j\omega\tau}) d\omega,$$

where $(a)$ results from expressing $\mathcal{X}$ by its spectral representation [cf. (25)] and $(b)$ from recalling (24). As in (25), the filter output can be also written as $\mathcal{Y} = \frac{1}{2\pi}\sum_{i=1}^{N}\int_{0}^{\infty}\tilde{\mathcal{Y}}(\lambda_i, j\omega)(\mathbf{v}_i \otimes e^{j\omega\tau})d\omega$, and thus we have $\tilde{\mathcal{Y}}(\lambda_i, j\omega) = \tilde{\mathcal{X}}(\lambda_i, j\omega)\int_{0}^{\infty}h(t)e^{-t(\lambda_i+j\omega)}dt$, following from (26). From the convolution theorem, the convolution operator implies a multiplication in the spectral domain. Therefore, the *frequency response* of the space-time graph filter is

$$\tilde{h}(\lambda, j\omega) = \int_{0}^{\infty}h(t)e^{-t(\lambda+j\omega)}dt, \tag{27}$$

which is the Laplace transform of the impulse response $h(t)$. It is worth noting that the filter response depends on the filter coefficients, which is irrespective of the graph. The spectral coefficients of the filter applied to one specific graph are, however, obtained by instantiating the frequency response $\tilde{h}(\lambda, j\omega)$ on its eigenvalues $\{\lambda_i\}_{i=1}^{N}$.

## B  PROOF OF PROPOSITION 1

First, we define the eigenvector misalignment between $\mathbf{S}$ and $\mathbf{E}$ in the following lemma (Gama et al., 2020a).

**Lemma 1.** *Let $\mathbf{S} = \mathbf{V}\boldsymbol{\Lambda}\mathbf{V}^{H}$ and $\mathbf{E} = \mathbf{U}\mathbf{M}\mathbf{U}^{H}$ with $\|\mathbf{E}\| \leq \epsilon_s$. For any eigenvector $\mathbf{v}_i$ of $\mathbf{S}$, it holds that*

$$\mathbf{E}\mathbf{v}_i = m_i\mathbf{v}_i + \mathbf{E_i}\mathbf{v}_i, \tag{28}$$

*with $\|\mathbf{E}_i\| \leq \epsilon\delta$, where $\delta = (\|\mathbf{U} - \mathbf{V}\| + 1)^2 - 1$.*

Now we proceed with the proof of Proposition 1.

*Proof of Proposition 1.* Proposition 1 bounds the distance between the space-time graph filters before and after adding graph perturbations. From (12), this distance can be evaluated as

$$\begin{aligned}
\|\mathbf{H}(\mathbf{S},\mathcal{L}_\tau) - \mathbf{H}(\hat{\mathbf{S}},\mathcal{L}_\tau)\|_{\mathcal{P}} &= \min_{\mathbf{P}\in\mathcal{P}}\max_{\mathcal{X}:\|\mathcal{X}\|_F=1}\|\mathbf{H}(\mathbf{S},\mathcal{L}_\tau)\mathcal{X} - \mathbf{P}^T\mathbf{H}(\hat{\mathbf{S}},\mathcal{L}_\tau)\mathbf{P}\mathcal{X}\|_F \\
&= \min_{\mathbf{P}\in\mathcal{P}}\max_{\mathcal{X}:\|\mathcal{X}\|_F=1}\|\mathbf{H}(\mathbf{S},\mathcal{L}_\tau)\mathcal{X} - \mathbf{H}(\mathbf{P}^T\hat{\mathbf{S}}\mathbf{P},\mathcal{L}_\tau)\mathcal{X}\|_F,
\end{aligned} \tag{29}$$

where the last step is due to $\mathbf{H}$ being permutation equivariant, i.e.,

$$\mathbf{P}^T\mathbf{H}(\mathbf{S},\mathcal{L}_\tau)\mathbf{P} = \int_0^\infty h(t)\mathbf{P}^T e^{-t\mathbf{S}\circ\mathcal{L}_\tau}\mathbf{P}dt \overset{(a)}{=} \int_0^\infty h(t)e^{-t\mathbf{P}^T\mathbf{S}\mathbf{P}\circ\mathcal{L}_\tau} = \mathbf{H}(\mathbf{P}^T\mathbf{S}\mathbf{P},\mathcal{L}_\tau). \tag{30}$$

Equality $(a)$ results from the fact that matrix $\mathbf{P}$ commutes with the exponential operator and als with the TSO. From the minimum operator in (29), there exists a matrix $\mathbf{P}_0 \in \mathcal{P}$ that satisfies

$$\begin{aligned}
\|\mathbf{H}(\mathbf{S},\mathcal{L}_\tau) - \mathbf{H}(\hat{\mathbf{S}},\mathcal{L}_\tau)\|_{\mathcal{P}} &\leq \max_{\mathcal{X}:\|\mathcal{X}\|_F=1}\|\mathbf{H}(\mathbf{S},\mathcal{L}_\tau)\mathcal{X} - \mathbf{H}(\mathbf{P}_0^T\hat{\mathbf{S}}\mathbf{P}_0,\mathcal{L}_\tau)\mathcal{X}\|_F \\
&= \|\mathbf{H}(\mathbf{S},\mathcal{L}_\tau) - \mathbf{H}(\mathbf{P}_0^T\hat{\mathbf{S}}\mathbf{P}_0,\mathcal{L}_\tau)\| \\
&\overset{(b)}{=} \|\mathbf{H}(\mathbf{S},\mathcal{L}_\tau) - \mathbf{H}(\mathbf{S}+\mathbf{E}\mathbf{S}+\mathbf{S}\mathbf{E},\mathcal{L}_\tau)\|,
\end{aligned} \tag{31}$$

where $\|.\|$ is the operator norm, and $(b)$ follows from the assumption (A1).

Define the the filter difference as $\Delta_s := \mathbf{H}(\mathbf{S}+\mathbf{E}\mathbf{S}+\mathbf{S}\mathbf{E},\mathcal{L}_\tau) - \mathbf{H}(\mathbf{S},\mathcal{L}_\tau)$. The next step is to find the norm of $\Delta_s$. $\Delta_s$ can be expressed [cf. (5)] as

$$\Delta_s = \int_0^\infty h(t)e^{-t(\mathbf{S}+\mathbf{E}\mathbf{S}+\mathbf{S}\mathbf{E})\circ\mathcal{L}_\tau}dt - \int_0^\infty h(t)e^{-t\mathbf{S}\circ\mathcal{L}_\tau}dt. \tag{32}$$

From Taylor series, $e^{-t\mathbf{S}\circ\mathcal{L}_\tau} = \sum_{n=0}^{\infty}\frac{1}{n!}(-t)^n(\mathbf{S}\circ\mathcal{L}_\tau)^n$ and the difference is then written as

$$\Delta_s = \int_0^\infty h(t)\sum_{n=0}^{\infty}\frac{(-t)^n}{n!}\Big(((\mathbf{S}+\mathbf{E}\mathbf{S}+\mathbf{S}\mathbf{E})\circ\mathcal{L}_\tau)^n - (\mathbf{S}\circ\mathcal{L}_\tau)^n\Big)dt. \tag{33}$$

By induction, we expand the term $((\mathbf{S} + \mathbf{ES} + \mathbf{SE}) \circ \mathcal{L}_\tau)^n$ to the first order on $\mathbf{E}$, so we have

$$
\begin{aligned}
((\mathbf{S} + \mathbf{ES} + \mathbf{SE}) \circ \mathcal{L}_\tau)^n &\overset{(a)}{=} (\mathbf{S} \circ \mathcal{L}_\tau + (\mathbf{ES} + \mathbf{SE}) \circ \mathcal{L}_\tau)^n \\
&= (\mathbf{S} \circ \mathcal{L}_\tau)^n + \sum_{r=0}^{n-1} (\mathbf{S} \circ \mathcal{L}_\tau)^r \Big( (\mathbf{ES} + \mathbf{SE}) \circ \mathcal{L}_\tau \Big) (\mathbf{S} \circ \mathcal{L}_\tau)^{n-r-1} + \mathbf{O}_2(\mathbf{E}),
\end{aligned}
\tag{34}
$$

where $\mathbf{O}_2(\mathbf{E})$ combines the other higher order terms, and $(a)$ follows from $\circ$ being a linear composition. Hence, (33) reduces to

$$
\Delta_s = \int_0^\infty h(t) \sum_{n=0}^\infty \frac{(-t)^n}{n!} \sum_{r=0}^{n-1} (\mathbf{S} \circ \mathcal{L}_\tau)^r \Big( (\mathbf{ES} + \mathbf{SE}) \circ \mathcal{L}_\tau \Big) (\mathbf{S} \circ \mathcal{L}_\tau)^{n-r-1} dt + \mathbf{O}(\mathbf{E}), \tag{35}
$$

with $\mathbf{O}(\mathbf{E}) = \int_0^\infty h(t) \sum_{n=0}^\infty \frac{(-t)^n}{n!} \mathbf{O}_2(\mathbf{E}) dt$. Note that the frequency response of the filter is an analytic function. Thus the norm of $\mathbf{O}(\mathbf{E})$ is of order $\mathcal{O}(\|\mathbf{E}\|^2)$ since

$$
0 < \lim_{\|\mathbf{E}\| \to 0} \frac{\|\mathbf{O}(\mathbf{E})\|}{\|\mathbf{E}\|^2} < \infty. \tag{36}
$$

Denote the first term in the right hand side of (35) by $\Delta(\mathbf{S})$. After splitting the summands $(\mathbf{ES} + \mathbf{SE}) \circ \mathcal{L}_\tau$ in (35) into $\mathbf{ES} \circ \mathcal{L}_\tau + \mathbf{SE} \circ \mathcal{L}_\tau$, we get

$$
\begin{aligned}
\Delta(\mathbf{S}) = &\int_0^\infty h(t) \sum_{n=0}^\infty \frac{(-t)^n}{n!} \sum_{r=0}^{n-1} (\mathbf{S} \circ \mathcal{L}_\tau)^r \mathbf{E} (\mathbf{S} \circ \mathcal{L}_\tau)^{n-r} dt \\
+ &\int_0^\infty h(t) \sum_{n=0}^\infty \frac{(-t)^n}{n!} \sum_{r=0}^{n-1} (\mathbf{S} \circ \mathcal{L}_\tau)^{r+1} \mathbf{E} (\mathbf{S} \circ \mathcal{L}_\tau)^{n-r-1} dt.
\end{aligned}
\tag{37}
$$

The first line is straightforward and the second follows from commuting the matrix $\mathbf{E}$ with the TSO, $\mathcal{L}_\tau$. The required norm becomes $\|\Delta_s\| \le \|\Delta(\mathbf{S})\| + \|\mathbf{O}(\mathbf{E})\|$ from the triangle inequality, with $\|\mathbf{O}(\mathbf{E})\|$ being of order $\mathcal{O}(\epsilon_s^2)$ by the assumption (A2) and (36). Thus the aim of the proof reduces to bound the norm $\|\Delta(\mathbf{S})\|$ by $2C\epsilon_s(1 + \delta\sqrt{N})$.

The norm $\|\Delta(\mathbf{S})\|$ is defined as $\|\Delta(\mathbf{S})\| = \max_{\mathcal{X}:\|\mathcal{X}\|_F=1} \|\Delta(\mathbf{S})\mathcal{X}\|_F$. We express $\Delta(\mathbf{S})\mathcal{X}$ as

$$
\begin{aligned}
\Delta(\mathbf{S})\mathcal{X} = &\frac{1}{2\pi} \sum_{i=1}^N \int_0^\infty \tilde{\mathcal{X}}(\lambda_i, j\omega) \int_0^\infty h(t) \sum_{n=0}^\infty \frac{(-t)^n}{n!} \sum_{r=0}^{n-1} (\mathbf{S} \circ \mathcal{L}_\tau)^r \mathbf{E} (\mathbf{S} \circ \mathcal{L}_\tau)^{n-r} (\mathbf{v}_i \otimes e^{j\omega\tau}) dt d\omega \\
+ &\frac{1}{2\pi} \sum_{i=1}^N \int_0^\infty \tilde{\mathcal{X}}(\lambda_i, j\omega) \int_0^\infty h(t) \sum_{n=0}^\infty \frac{(-t)^n}{n!} \sum_{r=0}^{n-1} (\mathbf{S} \circ \mathcal{L}_\tau)^{r+1} \mathbf{E} (\mathbf{S} \circ \mathcal{L}_\tau)^{n-r-1} (\mathbf{v}_i \otimes e^{j\omega\tau}) dt d\omega,
\end{aligned}
\tag{38}
$$

which follows from expressing $\Delta(\mathbf{S})$ as in (37) and $\mathcal{X}$ as in (25). Then

$$
\begin{aligned}
\Delta(\mathbf{S})\mathcal{X} = &\frac{1}{2\pi} \sum_{i=1}^N \int_0^\infty \tilde{\mathcal{X}}(\lambda_i, j\omega) \int_0^\infty h(t) \sum_{n=0}^\infty \frac{(-t)^n}{n!} \sum_{r=0}^{n-1} (\lambda_i + j\omega)^{n-r} (\mathbf{S} \circ \mathcal{L}_\tau)^r \mathbf{E} (\mathbf{v}_i \otimes e^{j\omega\tau}) dt d\omega \\
+ &\frac{1}{2\pi} \sum_{i=1}^N \int_0^\infty \tilde{\mathcal{X}}(\lambda_i, j\omega) \int_0^\infty h(t) \sum_{n=0}^\infty \frac{(-t)^n}{n!} \sum_{r=0}^{n-1} (\lambda_i + j\omega)^{n-r-1} (\mathbf{S} \circ \mathcal{L}_\tau)^{r+1} \mathbf{E} (\mathbf{v}_i \otimes e^{j\omega\tau}) dt d\omega,
\end{aligned}
\tag{39}
$$

since $(\mathbf{S} \circ \mathcal{L}_\tau)^q (\mathbf{v}_i \otimes e^{j\omega\tau}) = (\lambda_i + j\omega)^q (\mathbf{v}_i \otimes e^{j\omega\tau})$. The assumption (A2) indicates that matrix $\mathbf{E}$ has eigenvectors that are not aligned with $\mathbf{v}_i$ $\forall i$. From Lemma 1, it follows that $\mathbf{E}(\mathbf{v}_i \otimes e^{j\omega\tau}) = \mathbf{E}\mathbf{v}_i \otimes e^{j\omega\tau} = m_i \mathbf{v}_i \otimes e^{j\omega\tau} + \mathbf{E}_i \mathbf{v}_i \otimes e^{j\omega\tau}$. The terms $(\mathbf{S} \circ \mathcal{L}_\tau)^p \mathbf{E}(\mathbf{v}_i \otimes e^{j\omega\tau})$ of (39) can then be simplified as

$$
\begin{aligned}
(\mathbf{S} \circ \mathcal{L}_\tau)^p \mathbf{E}(\mathbf{v}_i \otimes e^{j\omega\tau}) &= m_i (\mathbf{S} \circ \mathcal{L}_\tau)^p (\mathbf{v}_i \otimes e^{j\omega\tau}) + (\mathbf{S} \circ \mathcal{L}_\tau)^p \mathbf{E}_i (\mathbf{v}_i \otimes e^{j\omega\tau}) \\
&= m_i (\lambda_i + j\omega)^p (\mathbf{v}_i \otimes e^{j\omega\tau}) + (\mathbf{S} \circ \mathcal{L}_\tau)^p \mathbf{E}_i (\mathbf{v}_i \otimes e^{j\omega\tau}).
\end{aligned}
\tag{40}
$$

Substituting the first term of (40) in (39) results in

$$\Delta_1(\mathbf{S})\mathcal{X} := \frac{1}{2\pi}\sum_{i=1}^{N}\int_0^{\infty}\tilde{\mathcal{X}}(\lambda_i, j\omega)\int_0^{\infty}h(t)\sum_{n=0}^{\infty}\frac{(-t)^n}{n!}\sum_{r=0}^{n-1}2m_i(\lambda_i+j\omega)^n(\mathbf{v}_i\otimes e^{j\omega\tau})dtd\omega.$$

(41)

Doing the same for the second term of (40) leads to

$$\Delta_2(\mathbf{S})\mathcal{X} := \frac{1}{2\pi}\sum_{i=1}^{N}\int_0^{\infty}\tilde{\mathcal{X}}(\lambda_i, j\omega)\int_0^{\infty}h(t)\sum_{n=0}^{\infty}\frac{(-t)^n}{n!}\sum_{r=0}^{n-1}(\lambda_i+j\omega)^{n-r}(\mathbf{S}\circ\mathcal{L}_\tau)^r\mathbf{E}_i(\mathbf{v}_i\otimes e^{j\omega\tau})dtd\omega$$
$$+\frac{1}{2\pi}\sum_{i=1}^{N}\int_0^{\infty}\tilde{\mathcal{X}}(\lambda_i, j\omega)\int_0^{\infty}h(t)\sum_{n=0}^{\infty}\frac{(-t)^n}{n!}\sum_{r=0}^{n-1}(\lambda_i+j\omega)^{n-r-1}(\mathbf{S}\circ\mathcal{L}_\tau)^{r+1}\mathbf{E}_i(\mathbf{v}_i\otimes e^{j\omega\tau})dtd\omega,$$

(42)

where $\Delta(\mathbf{S})\mathcal{X} = \Delta_1(\mathbf{S})\mathcal{X} + \Delta_2(\mathbf{S})\mathcal{X}$. The next step is to find the norm of $\Delta_1(\mathbf{S})$ and $\Delta_2(\mathbf{S})$.

For the term $\Delta_1(\mathbf{S})\mathcal{X}$ in (41), we notice that the inner summation no longer depends on $r$ and can be written as $2nm_i(\lambda_i+j\omega)^n(\mathbf{v}_i\otimes e^{j\omega\tau})$. Therefore, we get

$$2m_i\int_0^{\infty}h(t)\sum_{n=0}^{\infty}n\frac{(-t)^n}{n!}(\lambda_i+j\omega)^n dt = 2m_i(\lambda_i+j\omega)\frac{\partial}{\partial\lambda}\tilde{h}(\lambda_i, j\omega),$$

(43)

where $\frac{\partial}{\partial\lambda}\tilde{h}(\lambda_i, j\omega) = \int_0^{\infty}h(t)\sum_{n=0}^{\infty}\frac{n}{n!}(-t)^n(\lambda_i+j\omega)^{n-1}dt$ is the partial derivative of the frequency response of the filter computed at $\lambda = \lambda_i$. Thus (41) is written as

$$\Delta_1(\mathbf{S})\mathcal{X} = \frac{1}{2\pi}\sum_{i=1}^{N}\int_0^{\infty}2m_i\tilde{\mathcal{X}}(\lambda_i, j\omega)(\lambda_i+j\omega)\frac{\partial}{\partial\lambda}\tilde{h}(\lambda_i, j\omega)(\mathbf{v}_i\otimes e^{j\omega\tau})d\omega.$$

(44)

From Pythagoras' theorem we know that the squared norm of a sum of orthogonal terms is the sum of the squared norm of the individual summands. Then

$$\|\Delta_1(\mathbf{S})\mathcal{X}\|_F^2 = \sum_{i=1}^{N}\int_0^{\infty}4m_i^2|\tilde{\mathcal{X}}(\lambda_i, j\omega)|^2\cdot|\lambda_i+j\omega|^2\cdot\left|\frac{\partial}{\partial\lambda}\tilde{h}(\lambda_i, j\omega)\right|^2 d\omega \leq 4\epsilon_s^2 C^2\|\mathcal{X}\|_F^2,$$ (45)

since $|m_i| \leq \|\mathbf{E}\| \leq \epsilon_s$, the filter is integral Lipschitz [cf. (17)], and the eigenfunction basis vectors are orthonormal, i.e., $\|\mathbf{v}_k\otimes\frac{1}{2\pi}e^{j\omega_*\tau}\|_F^2 = 1$. The norm $\|\mathcal{X}\|_F^2$ is defined as $\sum_{i=1}^{N}\int_0^{\infty}|\tilde{\mathcal{X}}(\lambda_i, j\omega)|^2 d\omega$.

We now rewritte $\Delta_2(\mathbf{S})\mathcal{X}$ in (42) as

$$\Delta_2(\mathbf{S})\mathcal{X} = \frac{1}{2\pi}\sum_{i=1}^{N}\int_0^{\infty}\tilde{\mathcal{X}}(\lambda_i, j\omega)\mathcal{K}_i\mathbf{E}_i(\mathbf{v}_i\otimes e^{j\omega\tau})d\omega,$$

(46)

with

$$\mathcal{K}_i = \int_0^{\infty}h(t)\sum_{n=0}^{\infty}\frac{(-t)^n}{n!}\sum_{r=0}^{n-1}\left((\lambda_i+j\omega)^{n-r}(\mathbf{S}\circ\mathcal{L}_\tau)^r + (\lambda_i+j\omega)^{n-r-1}(\mathbf{S}\circ\mathcal{L}_\tau)^{r+1}\right)dt.$$ (47)

We next find the norm of $\mathcal{K}_i$ in order to find $\|\Delta_2(\mathbf{S})\|$. Note that $\mathcal{K}_i : L^2(\mathbb{R}^N)\otimes L^2(\mathbb{R}) \to L^2(\mathbb{R}^N)\otimes L^2(\mathbb{R})$ is a linear operator and so is the matrix $\mathbf{S} : L^2(\mathbb{R}^N) \to L^2(\mathbb{R}^N)$. A matrix $\mathbf{S}$ can be expressed using the outer product of its eigenvectors as $\mathbf{S} = \sum_{i=1}^{N}\lambda_i\mathbf{v}_i\mathbf{v}_i^H$. Since the tensor product generalizes the outer product of n-dimensional vectors, we can draw a parallel and have

$$(\mathbf{S}\circ\mathcal{L}_\tau)^q = \frac{1}{(2\pi)^2}\sum_{k=1}^{N}\int_0^{\infty}(\lambda_k+j\omega_*)^q(\mathbf{v}_k\otimes e^{j\omega_*\tau})\otimes(\mathbf{v}_k\otimes e^{j\omega_*\tau})^H d\omega_*.$$

(48)

Calculating (47) requires evaluating the term $(\lambda_i + j\omega)^{n-r}(\mathbf{S} \circ \mathcal{L}_\tau)^r + (\lambda_i + j\omega)^{n-r-1}(\mathbf{S} \circ \mathcal{L}_\tau)^{r+1}$ first, which can be simplified as

$$(\lambda_i + j\omega)^{n-r}(\mathbf{S} \circ \mathcal{L}_\tau)^r + (\lambda_i + j\omega)^{n-r-1}(\mathbf{S} \circ \mathcal{L}_\tau)^{r+1}$$
$$= (\lambda_i + j\omega)^n \left( \frac{(\mathbf{S} \circ \mathcal{L}_\tau)^r}{(\lambda_i + j\omega)^r} + \frac{(\mathbf{S} \circ \mathcal{L}_\tau)^{r+1}}{(\lambda_i + j\omega)^{r+1}} \right). \tag{49}$$

When we substitute (48) in (49), we get

$$(\lambda_i + j\omega)^{n-r}(\mathbf{S} \circ \mathcal{L}_\tau)^r + (\lambda_i + j\omega)^{n-r-1}(\mathbf{S} \circ \mathcal{L}_\tau)^{r+1}$$
$$= \frac{(\lambda_i + j\omega)^n}{(2\pi)^2} \sum_{k=1}^{N} \int_0^\infty \left( \frac{(\lambda_k + j\omega_*)^r}{(\lambda_i + j\omega)^r} + \frac{(\lambda_k + j\omega_*)^{r+1}}{(\lambda_i + j\omega)^{r+1}} \right) (\mathbf{v}_k \otimes e^{j\omega_* \tau}) \otimes (\mathbf{v}_k \otimes e^{j\omega_* \tau})^H d\omega_*. \tag{50}$$

The inner summation in (47) reduces to the sum of two geometric series, each of which has the form

$$\sum_{r=0}^{n-1} \left( \frac{\lambda_k + j\omega_*}{\lambda_i + j\omega} \right)^r = \frac{1}{(\lambda_i + j\omega)^{n-1}} \frac{(\lambda_i + j\omega)^n - (\lambda_k + j\omega_*)^n}{\lambda_i + j\omega - (\lambda_k + j\omega_*)}, \tag{51}$$

and the reader can confirm with a simple algebraic manipulation that the right hand side of (51) follows from the geometric sum $\sum_{r=0}^{n-1} a^r = (1 - a^n)/(1 - a)$. It is also straightforward to show that

$$(\lambda_i + j\omega)^n \sum_{r=0}^{n-1} \left( \frac{\lambda_k + j\omega_*}{\lambda_i + j\omega} \right)^r + \left( \frac{\lambda_k + j\omega_*}{\lambda_i + j\omega} \right)^{r+1} = \frac{\lambda_i + \lambda_k + j(\omega + \omega_*)}{\lambda_i - \lambda_k + j(\omega - \omega_*)} \left( (\lambda_i + j\omega)^n - (\lambda_k + j\omega_*)^n \right). \tag{52}$$

with some algebraic manipulations. We eventually can write $\mathcal{K}_i$ in (47) as

$$\mathcal{K}_i = \int_0^\infty h(t) \sum_{n=0}^\infty \frac{(-t)^n}{(2\pi)^2 \, n!} \sum_{k=1}^N \int_0^\infty \frac{\lambda_i + \lambda_k + j(\omega + \omega_*)}{\lambda_i - \lambda_k + j(\omega - \omega_*)} \left( (\lambda_i + j\omega)^n - (\lambda_k + j\omega_*)^n \right) \tag{53}$$
$$(\mathbf{v}_k \otimes e^{j\omega_* \tau}) \otimes (\mathbf{v}_k \otimes e^{j\omega_* \tau})^H d\omega_* dt.$$

Since we have $\tilde{h}(\lambda_i, j\omega) = \int_0^\infty h(t) e^{-t(\lambda_i + j\omega)} dt = \int_0^\infty h(t) \sum_{n=0}^\infty \frac{1}{n!}(-t)^n (\lambda_i + j\omega)^n dt$ from Taylor series, $\mathcal{K}_i$ reduces to

$$\mathcal{K}_i = \frac{1}{(2\pi)^2} \sum_{k=1}^N \int_0^\infty \frac{(\lambda_i + j\omega) + (\lambda_k + j\omega_*)}{(\lambda_i + j\omega) - (\lambda_k - j\omega_*)} \left( \tilde{h}(\lambda_i, j\omega) - \tilde{h}(\lambda_k, j\omega_*) \right) \tag{54}$$
$$(\mathbf{v}_k \otimes e^{j\omega_* \tau}) \otimes (\mathbf{v}_k \otimes e^{j\omega_* \tau})^H d\omega_*.$$

Note that (54) has the form $(\mathbf{w}_2 + \mathbf{w}_1)(\tilde{h}(\mathbf{w}_2) - \tilde{h}(\mathbf{w}_1))/(\mathbf{w}_2 - \mathbf{w}_1)$ of (16) with $\mathbf{w}$ being replaced by its complex form, $\lambda + j\omega$. This remark will help bound the norm of $\mathcal{K}_i$. The norm $\|\mathcal{K}_i\|$ is defined as $\max_{\mathcal{X}: \|\mathcal{X}\|_F = 1} \|\mathcal{K}_i \mathcal{X}\|_F$ for all $i$, and we have

$$\mathcal{K}_i \mathcal{X} \overset{(a)}{=} \mathcal{K}_i \left( \frac{1}{2\pi} \sum_{m=1}^N \int_0^\infty \tilde{\mathcal{X}}(\lambda_m, j\dot{\omega})(\mathbf{v}_m \otimes e^{j\dot{\omega}\tau}) d\dot{\omega} \right)$$
$$\overset{(b)}{=} \frac{1}{2\pi} \sum_{k=1}^N \int_0^\infty \tilde{\mathcal{X}}(\lambda_k, j\omega_*) \frac{(\lambda_i + j\omega) + (\lambda_k + j\omega_*)}{(\lambda_i + j\omega) - (\lambda_k - j\omega_*)} \left( \tilde{h}(\lambda_i, j\omega) - \tilde{h}(\lambda_k, j\omega_*) \right) (\mathbf{v}_k \otimes e^{j\omega_* \tau}) d\omega_*. \tag{55}$$

The RHS in $(a)$ involves the inner products $\frac{1}{(2\pi)^2}(\mathbf{v}_k \otimes e^{j\omega_* \tau})^H (\mathbf{v}_m \otimes e^{j\omega \tau}) = \frac{1}{(2\pi)^2} \langle \mathbf{v}_k \otimes e^{j\omega_* \tau}, \mathbf{v}_m \otimes e^{j\omega \tau} \rangle$, $\forall m, k$. These inner products are between the orthogonal eigenvectors and are nonzero if and only if $m = k$ and $\dot{\omega} = \omega_*$. Therefore, only the terms that have $m = k$ and $\dot{\omega} = \omega_*$ appeared in $(b)$, and the inner products are 1.

From Pythagoras' theorem, the squared norm of $\mathcal{K}_i \mathcal{X}$ is then

$$
\begin{aligned}
\|\mathcal{K}_i \mathcal{X}\|_F^2 &= \sum_{k=1}^{N} \int_0^\infty |\tilde{\mathcal{X}}(\lambda_k, j\omega_*)|^2 \left| \frac{(\lambda_i + j\omega) + (\lambda_k + j\omega_*)}{(\lambda_i + j\omega) - (\lambda_k - j\omega_*)} \left( \tilde{h}(\lambda_i, j\omega) - \tilde{h}(\lambda_k, j\omega_*) \right) \right|^2 d\omega_* \\
&\leq 4C^2 \|\mathcal{X}\|_F^2, \ \forall i,
\end{aligned}
\tag{56}
$$

followed from the assumption (A3) that the filter is Lipschitz filter [cf. Definition 4]. Therefore, we get $\|\mathcal{K}_i\| \leq 2C$ for a unit-norm signal $\mathcal{X}$. Now, it is left to take the norm of (46), which yields

$$
\begin{aligned}
\|\Delta_2(\mathbf{S})\mathcal{X}\|_F &= \left\| \frac{1}{2\pi} \sum_{i=1}^{N} \int_0^\infty \tilde{\mathcal{X}}(\lambda_i, j\omega) \mathcal{K}_i \mathbf{E}_i (\mathbf{v}_i \otimes e^{j\omega\tau}) d\omega \right\|_F \\
&\leq \sum_{i=1}^{N} \int_0^\infty |\tilde{X}(\lambda_i, j\omega)| \cdot \|\mathcal{K}_i\| \cdot \|\mathbf{E}_i\| d\omega \leq 2C\epsilon\delta \sum_{i=1}^{N} \int_0^\infty |\tilde{X}(\lambda_i, j\omega)| d\omega,
\end{aligned}
\tag{57}
$$

where we have $\|\mathbf{E}_i\| \leq \epsilon\delta, \ \forall i$ from Lemma 1. To bound the summation in (57), we can write

$$
\begin{aligned}
\sum_{i=1}^{N} \int_0^\infty |\tilde{\mathcal{X}}(\lambda_i, j\omega)| \cdot 1 \, d\omega &\overset{(a)}{\leq} \left( \sum_{i=1}^{N} \int_0^\infty |\tilde{\mathcal{X}}(\lambda_i, j\omega)|^2 d\omega \right)^{\frac{1}{2}} \left( \sum_{i=1}^{N} \int_0^\infty 1 \, d\omega \right)^{\frac{1}{2}} \\
&\overset{(b)}{=} \left( \sum_{i=1}^{N} \int_0^\infty |\tilde{\mathcal{X}}(\lambda_i, j\omega)|^2 d\omega \right)^{\frac{1}{2}} \left( \sum_{i=1}^{N} \int_0^\infty \hat{\delta}(\tau) d\tau \right)^{\frac{1}{2}} = \sqrt{N} \|\mathcal{X}\|_F,
\end{aligned}
\tag{58}
$$

where $(a)$ follows from Cauchy-Schwartz inequality, and $(b)$ from Parseval's theorem. Note that $\hat{\delta}(\tau)$ is the Dirac delta function. Eventually, we obtain

$$
\|\Delta_2(\mathbf{S})\mathcal{X}\|_F \leq 2C\epsilon_s\delta\sqrt{N}\|\mathcal{X}\|_F.
\tag{59}
$$

Recall that $\Delta(\mathbf{S})\mathcal{X} = \Delta_1(\mathbf{S})\mathcal{X} + \Delta_2(\mathbf{S})\mathcal{X}$. From the triangle inequality, we then obtain

$$
\|\Delta(\mathbf{S})\mathcal{X}\|_F \leq \|\Delta_1(\mathbf{S})\mathcal{X}\|_F + \|\Delta_2(\mathbf{S})\mathcal{X}\|_F \leq 2C\epsilon_s(1 + \delta\sqrt{N})\|\mathcal{X}\|_F.
\tag{60}
$$

following from and using (45) and (59). Hence, $\|\Delta(\mathbf{S})\| = \max_{\mathcal{X}:\|\mathcal{X}\|_F=1} |\Delta(\mathbf{S})\mathcal{X}|_F \leq 2C\epsilon_s(1 + \delta\sqrt{N})$, which completes the proof.

## C    PROOF OF PROPOSITION 2

The aim of the proof is to bound the distance between the space-time graph filters before and after adding time perturbation. This distance is calculated using the operator distance modulo translation. From Definition 3, it holds that there exists a real value $s$ such that

$$
\begin{aligned}
\|\mathbf{H}(\mathbf{S}, \mathcal{L}_\tau) - \mathbf{H}(\mathbf{S}, \hat{\mathcal{L}}_\tau)\|_{\mathcal{T}} &= \min_{s\in\mathbb{R}} \max_{x:\|x\|_F=1} \left\| \mathbf{H}(\mathbf{S}, \mathcal{L}_\tau)\mathcal{X} - e^{-s\mathcal{L}_\tau} \mathbf{H}(\mathbf{S}, \hat{\mathcal{L}}_\tau)\mathcal{X} \right\|_F \\
&\overset{(a)}{\leq} \max_{\mathcal{X}:\|\mathcal{X}\|_F=1} \|\mathbf{H}(\mathbf{S}, \mathcal{L}_\tau)\mathcal{X} - \mathbf{H}(\mathbf{S}, \hat{\mathcal{L}}_\tau)\mathcal{X}\|_F \\
&\overset{(b)}{=} \|\mathbf{H}(\mathbf{S}, \mathcal{L}_\tau) - \mathbf{H}(\mathbf{S}, (1 + \xi(\tau))\mathcal{L}_\tau)\|,
\end{aligned}
\tag{61}
$$

where in $(a)$ we let the minimum norm value be lower than or equal to the norm when $s = 0$, and in $(b)$ we use assumption (A1). The difference between the filters is evaluated as

$$
\mathbf{H}(\mathbf{S}, \hat{\mathcal{L}}_\tau) - \mathbf{H}(\mathbf{S}, \mathcal{L}_\tau) = \int_0^\infty h(t) e^{-t\mathbf{S}\circ(\mathcal{L}_\tau + \xi(\tau)\mathcal{L}_\tau)} dt - \int_0^\infty h(t) e^{-t\mathbf{S}\circ\mathcal{L}_\tau} dt.
\tag{62}
$$

Let $\Delta_\tau$ denote $\mathbf{H}(\mathbf{S}, \hat{\mathcal{L}}_\tau) - \mathbf{H}(\mathbf{S}, \mathcal{L}_\tau)$, which can be written as

$$
\Delta_\tau = \int_0^\infty h(t) \sum_{n=0}^{\infty} \frac{(-t)^n}{n!} \left( (\mathbf{S} \circ (\mathcal{L}_\tau + \xi(\tau)\mathcal{L}_\tau))^n - (\mathbf{S} \circ \mathcal{L}_\tau)^n \right) dt,
\tag{63}
$$

following from expanding the exponentials to $e^{-t\mathbf{S}\circ\mathcal{L}_\tau} = \sum_{n=0}^{\infty} \frac{1}{n!}(-t)^n(\mathbf{S}\circ\mathcal{L}_\tau)^n$ using Taylor series. We also expand $(\mathbf{S}\circ\mathcal{L}_\tau + \mathbf{S}\circ\xi(\tau)\mathcal{L}_\tau)^n$ to the first order of $\xi(\tau)$:

$$(\mathbf{S}\circ\mathcal{L}_\tau + \mathbf{S}\circ\xi(\tau)\mathcal{L}_\tau)^n = (\mathbf{S}\circ\mathcal{L}_\tau)^n + \sum_{r=0}^{n-1}(\mathbf{S}\circ\mathcal{L}_\tau)^r\xi(\tau)(\mathbf{S}\circ\mathcal{L}_\tau)^{n-r} + \mathbf{O}_2(\xi(\tau)), \quad (64)$$

where $\mathbf{O}_2(\xi(\tau))$ is a polynomial of the higher powers of $\xi(\tau)$. By substituting (64) in (63), we can re-write the latter as

$$\Delta_\tau = \int_0^\infty h(t)\sum_{n=0}^{\infty}\frac{(-t)^n}{n!}\sum_{r=0}^{n-1}(\mathbf{S}\circ\mathcal{L}_\tau)^r\xi(\tau)(\mathbf{S}\circ\mathcal{L}_\tau)^{n-r}dt + \mathbf{O}(\xi(\tau)), \quad (65)$$

where $\mathbf{O}(\xi(\tau)) = \int_0^\infty h(t)\sum_{n=0}^{\infty}\frac{(-t)^n}{n!}\mathbf{O}_2(\xi(\tau))dt$. The quantity $\mathbf{O}(\xi(\tau))$ has all the higher power terms and satisfies

$$0 < \lim_{\|\xi(\tau)\|\to 0}\frac{\|\mathbf{O}(\xi(\tau))\|_2}{\|\xi(\tau)\|_2^2} < \infty. \quad (66)$$

Therefore, the norm $\|\mathbf{O}(\xi(\tau))\|_2$ is of order $\mathcal{O}(\epsilon_\tau^2)$ following from assumption (A2). Remember that if the norm is of order $\epsilon_\tau$, it means that $\|\xi(\tau)\|_2 \le \kappa\epsilon_\tau$ with $\kappa$ being an absolute constant. Our goal reduces to bound the norm of the first term of the right hand side of (65).

Since the TSO and GSO can commute, it holds that

$$(\mathbf{S}\circ\mathcal{L}_\tau)^r\xi(\tau)(\mathbf{S}\circ\mathcal{L}_\tau)^{n-r} = (\mathbf{S}^r\circ\mathcal{L}_\tau^r)\xi(\tau)(\mathbf{S}^{n-r}\circ\mathcal{L}_\tau^{n-r}) = \mathbf{S}^n\circ\mathcal{L}_\tau^r\xi(\tau)\mathcal{L}_\tau^{n-r}. \quad (67)$$

However, $\mathcal{L}_\tau$ and $\xi(\tau)$ do not commute due to their dependence on $\tau$. Recalling that $\mathcal{L}_\tau$ is a differential operator and applying the chain rule, we obtain

$$\mathcal{L}_\tau^r\left(\xi(\tau)\mathcal{L}_\tau^{n-r}\right) \stackrel{(a)}{=} \sum_{m=0}^{r}\binom{r}{m}\xi^{(m)}(\tau)\mathcal{L}_\tau^{n-m} = \xi(\tau)\mathcal{L}_\tau^n + \mathbf{G}_2(\xi'(\tau)), \quad (68)$$

where $(a)$ is valid by induction, $\xi^{(m)}(\tau) = \partial^m\xi(\tau)/\partial\tau^m$ and the term $\mathbf{G}_2(\xi'(\tau))$ contains the higher-order derivatives starting from the first derivative. Substituting (67) and (68) in (65), we get

$$\Delta_\tau = \int_0^\infty h(t)\sum_{n=0}^{\infty}\frac{(-t)^n}{n!}\sum_{r=0}^{n-1}\mathbf{S}^n\circ\xi(\tau)\mathcal{L}_\tau^n dt + \mathbf{G}(\xi'(\tau)) + \mathbf{O}(\xi(\tau))$$

$$\stackrel{(a)}{=} \int_0^\infty h(t)\sum_{n=0}^{\infty}\frac{(-t)^n}{n!}n\xi(\tau)(\mathbf{S}\circ\mathcal{L}_\tau)^n dt + \mathbf{G}(\xi'(\tau)) + \mathbf{O}(\xi(\tau)). \quad (69)$$

The summands of the inner summation in (69) no longer depend on $r$ leading to the form in $(a)$. We also have $\mathbf{G}(\xi'(\tau)) = \int_0^\infty h(t)\sum_{n=0}^{\infty}\frac{(-t)^n}{n!}\sum_{r=0}^{n-1}\mathbf{G}_2(\xi'(\tau))dt$. It can now be shown that the norm $\|\mathbf{G}(\xi'(\tau))\|_2$ is of order $\mathcal{O}(\epsilon_\tau^2)$ since

$$0 < \lim_{\|\xi'(\tau)\|_2\to 0}\frac{\|\mathbf{G}(\xi'(\tau))\|_2}{\|\xi'(\tau)\|_2} < \infty, \quad (70)$$

and $\|\xi'(\tau)\|_2$ is of order $\epsilon_\tau^2$ according to assumption (A2).

Denote the first term in the right hand side of (69) by $\Delta(\mathcal{L}_\tau)$. Since both $\|\mathbf{O}(\xi(\tau))\|$ and $\|\mathbf{G}(\xi'(\tau))\|$ are of order $\epsilon_\tau^2$, the rest of the proof aims to show that the norm of $\Delta(\mathcal{L}_\tau)$ is bounded by $C\kappa\epsilon_\tau$. We have

$$\Delta(\mathcal{L}_\tau)\mathcal{X} = \left(\int_0^\infty h(t)\sum_{n=0}^{\infty}\frac{(-t)^n}{n!}n\xi(\tau)(\mathbf{S}\circ\mathcal{L}_\tau)^n dt\right)\left(\frac{1}{2\pi}\sum_{i=1}^{N}\int_0^\infty \tilde{\mathcal{X}}(\lambda_i, j\omega)(\mathbf{v}_i \otimes e^{j\omega\tau})d\omega\right)$$

$$= \frac{1}{2\pi}\sum_{i=1}^{N}\int_0^\infty \tilde{\mathcal{X}}(\lambda_i, j\omega)\int_0^\infty h(t)\sum_{n=0}^{\infty}\frac{(-t)^n}{n!}n\xi(\tau)(\mathbf{S}\circ\mathcal{L}_\tau)^n(\mathbf{v}_i \otimes e^{j\omega\tau})dtd\omega.$$

$$(71)$$

Recalling that $(\mathbf{v}_i \otimes e^{j\omega\tau})$ is an eigenfunction of $(\mathbf{S} \circ \mathcal{L}_\tau)$, we can re-write (71) as

$$
\Delta(\mathcal{L}_\tau)\mathcal{X} = \frac{1}{2\pi}\sum_{i=1}^{N}\int_{0}^{\infty}\tilde{\mathcal{X}}(\lambda_i, j\omega)\xi(\tau)\int_{0}^{\infty}h(t)\sum_{n=0}^{\infty}\frac{n}{n!}(-t)^n(\lambda_i+j\omega)^n dt (\mathbf{v}_i \otimes e^{j\omega\tau})d\omega
$$
$$
= \frac{1}{2\pi}\sum_{i=1}^{N}\int_{0}^{\infty}\tilde{\mathcal{X}}(\lambda_i, j\omega)\xi(\tau)(\lambda_i+j\omega)\frac{\partial}{\partial j\omega}\tilde{h}(\lambda_i, j\omega)(\mathbf{v}_i \otimes e^{j\omega\tau})d\omega,
$$
(72)

where $\frac{\partial}{\partial j\omega}\tilde{h}(\lambda_i, j\omega) = \int_{0}^{\infty}h(t)\sum_{n=0}^{\infty}\frac{n}{n!}(-t)^n(\lambda_i+j\omega)^{n-1}dt$. The the squared norm of (72) can then be evaluated as

$$
\|\Delta(\mathcal{L}_\tau)\mathcal{X}\|_F^2 \overset{(a)}{\leq} \int_{0}^{\infty}\sum_{i=1}^{N}\int_{0}^{\infty}\left|\tilde{\mathcal{X}}(\lambda_i, j\omega)\right|^2 |\xi(\tau)|^2 |\lambda_i+j\omega|^2 \left|\frac{\partial}{\partial j\omega}\tilde{h}(\lambda_i, j\omega)\right|^2 d\omega d\tau
$$
$$
= |\lambda_i+j\omega|^2 \left|\frac{\partial}{\partial j\omega}\tilde{h}(\lambda_i, j\omega)\right|^2 \left(\int_{0}^{\infty}|\xi(\tau)|^2 d\tau\right)\left(\int_{0}^{\infty}\sum_{i=1}^{N}\left|\tilde{\mathcal{X}}(\lambda_i, j\omega)\right|^2 d\omega\right)
$$
$$
= |\lambda_i+j\omega|^2 \left|\frac{\partial}{\partial j\omega}\tilde{h}(\lambda_i, j\omega)\right|^2 \|\xi(\tau)\|_2^2 \|\mathcal{X}\|_F^2 \overset{(b)}{\leq} C^2 \kappa^2 \epsilon_\tau^2 \|\mathcal{X}\|_F^2.
$$
(73)

In $(a)$, we use the inequality $|\int_0^\infty x dx|^2 \leq \int_0^\infty |x|^2 dx$. In $(b)$, we use assumption (A3), which states that the filter is integral Lipschitz with a constant $C$, and assumption (A2), which has the norm $\|\xi(\tau)\|_2$ bounded by $\kappa\epsilon_\tau$. Finally, the required norm can be calculated as $\|\Delta(\mathcal{L}\tau)\| = \max_{\mathcal{X}:\|\mathcal{X}\|_F=1}\|\Delta(\mathcal{L}\tau)\mathcal{X}\|_F \leq C\kappa\epsilon_\tau$, which completes the proof.

## D PROOF OF THEOREM 1

The aim of the proof is to bound the distance between the space-time graph filters before and after adding joint perturbations. From (15), this distance can be evaluated as

$$
\|\mathbf{H}(\mathbf{S}, \mathcal{L}_\tau) - \mathbf{H}(\hat{\mathbf{S}}, \hat{\mathcal{L}}_\tau)\|_{\mathcal{P},\mathcal{T}} = \min_{\mathbf{P}\in\mathcal{P}}\min_{s\in\mathbb{R}}\max_{\mathcal{X}:\|\mathcal{X}\|_F=1}\left\|\mathbf{H}(\mathbf{S}, \mathcal{L}_\tau)\mathcal{X} - \mathbf{H}\left(\mathbf{P}^T\hat{\mathbf{S}}\mathbf{P}, e^{-s\mathcal{L}_\tau}\hat{\mathcal{L}}_\tau\right)\mathcal{X}\right\|_F
$$
$$
\overset{(a)}{\leq} \max_{\mathcal{X}:\|\mathcal{X}\|_F=1}\left\|\mathbf{H}(\mathbf{S}, \mathcal{L}_\tau)\mathcal{X} - \mathbf{H}(\mathbf{S}+\mathbf{S}\mathbf{E}+\mathbf{E}\mathbf{S}, \hat{\mathcal{L}}_\tau)\mathcal{X}\right\|_F
$$
$$
= \left\|\mathbf{H}(\mathbf{S}, \mathcal{L}_\tau) - \mathbf{H}(\mathbf{S}+\mathbf{S}\mathbf{E}+\mathbf{E}\mathbf{S}, \hat{\mathcal{L}}_\tau)\right\|,
$$
(74)

where $(a)$ is true for any specified values of $\mathbf{P}$ and $s$, and we choose $\mathbf{P}_0$ and $0$ as in (31) and (61), respectively. Then, we add and subtract $\mathbf{H}(\mathbf{S}+\mathbf{S}\mathbf{E}+\mathbf{E}\mathbf{S}, \mathcal{L}_\tau)$ from the filter difference to get

$$
\mathbf{H}(\mathbf{S}, \mathcal{L}_\tau) - \mathbf{H}(\mathbf{S}+\mathbf{S}\mathbf{E}+\mathbf{E}\mathbf{S}, \hat{\mathcal{L}}_\tau) = \mathbf{H}(\mathbf{S}, \mathcal{L}_\tau) - \mathbf{H}(\mathbf{S}+\mathbf{S}\mathbf{E}+\mathbf{E}\mathbf{S}, \mathcal{L}_\tau)
$$
$$
+ \mathbf{H}(\mathbf{S}+\mathbf{S}\mathbf{E}+\mathbf{E}\mathbf{S}, \mathcal{L}_\tau) - \mathbf{H}(\mathbf{S}+\mathbf{S}\mathbf{E}+\mathbf{E}\mathbf{S}, \hat{\mathcal{L}}_\tau).
$$
(75)

Using the triangular inequality, the norm of the filter difference is bounded by

$$
\|\mathbf{H}(\mathbf{S}, \mathcal{L}_\tau) - \mathbf{H}(\mathbf{S}+\mathbf{S}\mathbf{E}+\mathbf{E}\mathbf{S}, \hat{\mathcal{L}}_\tau)\| \leq \|\mathbf{H}(\mathbf{S}, \mathcal{L}_\tau) - \mathbf{H}(\mathbf{S}+\mathbf{S}\mathbf{E}+\mathbf{E}\mathbf{S}, \mathcal{L}_\tau)\|
$$
$$
+ \|\mathbf{H}(\mathbf{S}+\mathbf{S}\mathbf{E}+\mathbf{E}\mathbf{S}, \mathcal{L}_\tau) - \mathbf{H}(\mathbf{S}+\mathbf{S}\mathbf{E}+\mathbf{E}\mathbf{S}, \hat{\mathcal{L}}_\tau)\|
$$
$$
= \|\Delta_s\| + \|\Delta_\tau\|.
$$
(76)

Note that in Proposition 2, $\Delta_\tau$ is defined for the GSO $\mathbf{S}$ but the proposition is valid for any GSO. The bounds of $\|\Delta_s\|$ and $\|\Delta_\tau\|$ are obtained in Propositions 1 and 2. Therefore, we get

$$
\|\mathbf{H}(\mathbf{S}, \mathcal{L}_\tau) - \mathbf{H}(\mathbf{S}+\mathbf{S}\mathbf{E}+\mathbf{E}\mathbf{S}, \hat{\mathcal{L}}_\tau)\| \leq 2C\epsilon_s(1+\delta\sqrt{N}) + C\kappa\epsilon_\tau + \mathcal{O}(\epsilon^2),
$$
(77)

where $\mathcal{O}(\epsilon^2)$ is the highest among $\mathcal{O}(\epsilon_s^2)$ and $\mathcal{O}(\epsilon_\tau^2)$. This completes the proof.

### D.1 FINITE-IMPULSE RESPONSE FILTERS

While Theorem 1 handles space-time graph filters with continuous-time impulse response $h(t)$, its results can be extended to finite-impulse response filters following the same proofs in Sections B, C, and D. In this context, FIR filters can be thought of as a special case of the filter $h(t)$, where we have a finite number of filter coefficients. We summarize this remark in the following lemma.

**Lemma 2.** *Define a space-time graph filter with a finite impulse response as*

$$\mathbf{H}_d(\mathbf{S}, \mathcal{L}_\tau) = \sum_{k=0}^{K} h_k e^{-kT_s \mathbf{S} \circ \mathcal{L}_\tau}. \tag{78}$$

*Under the GSOs defined in Proposition 1 and TSOs in Proposition 2, the distance between the FIR space-time graph filters $\mathbf{H}_d(\mathbf{S}, \mathcal{L}_\tau)$ and $\mathbf{H}_d(\hat{\mathbf{S}}, \hat{\mathcal{L}}_\tau)$ satisfies*

$$\|\mathbf{H}_d(\mathbf{S}, \mathcal{L}_\tau) - \mathbf{H}_d(\hat{\mathbf{S}}, \hat{\mathcal{L}}_\tau)\|_{\mathcal{P}, \mathcal{T}} \leq 2C\epsilon_s \left(1 + \delta\sqrt{N}\right) + C\kappa\epsilon_\tau + \mathcal{O}(\epsilon^2). \tag{79}$$

## E PROOF OF THEOREM 2

The goal of the proof is to show the difference between the GNN output before and after adding perturbations. The GNN output is the $L$ layer's output, i.e., $\mathcal{X}_L$. From (7), we can write the difference between the two GNNs as

$$\|\mathbf{\Phi}(\mathcal{X}_0, \mathcal{H}, \mathbf{S} \circ \mathcal{L}_\tau) - \mathbf{\Phi}(\mathcal{X}_0, \mathcal{H}, \hat{\mathbf{S}} \circ \hat{\mathcal{L}}_\tau)\|_F = \|\mathcal{X}_L - \hat{\mathcal{X}}_L\|_F$$
$$= \left\|\sigma\left(\mathbf{H}_L(\mathbf{S}, \mathcal{L}_\tau)\mathcal{X}_{L-1}\right) - \sigma\left(\mathbf{H}_L(\hat{\mathbf{S}}, \hat{\mathcal{L}}_\tau)\hat{\mathcal{X}}_{L-1}\right)\right\|_F, \tag{80}$$

where $\mathbf{H}_L(\mathbf{S}, \mathcal{L}_\tau)$ is the filter in (78) at Layer $L$, $\mathcal{X}_\ell$ is the output of layer $\ell$, and $\hat{\mathcal{X}}_\ell$ is the corresponding output after adding the perturbations.

From assumption (A2), we get $\|\sigma(\mathcal{X}_2) - \sigma(\mathcal{X}_1)\|_2 \leq \|\mathcal{X}_2 - \mathcal{X}_1\|_2$. Accordingly, the norm of the output difference at any layer $\ell \leq L$ is bounded by

$$\|\mathcal{X}_\ell - \hat{\mathcal{X}}_\ell\|_F \leq \left\|\mathbf{H}_\ell(\mathbf{S}, \mathcal{L}_\tau)\mathcal{X}_{\ell-1} - \mathbf{H}_\ell(\hat{\mathbf{S}}, \hat{\mathcal{L}}_\tau)\hat{\mathcal{X}}_{\ell-1}\right\|_F. \tag{81}$$

Add and subtract $\mathbf{H}_\ell(\hat{\mathbf{S}}, \hat{\mathcal{L}}_\tau)\mathcal{X}_{\ell-1}$ inside the norm in the right hand side. The norm of (81) can then be wriien as

$$\|\mathcal{X}_\ell - \hat{\mathcal{X}}_\ell\|_F \leq \left\|\mathbf{H}_\ell(\mathbf{S}, \mathcal{L}_\tau)\mathcal{X}_{\ell-1} - \mathbf{H}_\ell(\hat{\mathbf{S}}, \hat{\mathcal{L}}_\tau)\mathcal{X}_{\ell-1}\right\| + \left\|\mathbf{H}_\ell(\hat{\mathbf{S}}, \hat{\mathcal{L}}_\tau)\mathcal{X}_{\ell-1} - \mathbf{H}_\ell(\hat{\mathbf{S}}, \hat{\mathcal{L}}_\tau)\hat{\mathcal{X}}_{\ell-1}\right\|_F$$
$$\leq \left\|\mathbf{H}_\ell(\mathbf{S}, \mathcal{L}_\tau) - \mathbf{H}_\ell(\hat{\mathbf{S}}, \hat{\mathcal{L}}_\tau)\right\| \|\mathcal{X}_{\ell-1}\|_F + \left\|\mathbf{H}_\ell(\hat{\mathbf{S}}, \hat{\mathcal{L}}_\tau)\right\| \left\|\mathcal{X}_{\ell-1} - \hat{\mathcal{X}}_{\ell-1}\right\|_F, \tag{82}$$

using the triangle inequality in $(a)$ and Cauchy-Schwarz inequality in $(b)$. From assumption (A1), the filters have unit operator norm, i.e., $\|\mathbf{H}_\ell(\mathbf{S}, \mathcal{L}_\tau)\| = 1, \forall \ell = 1, \ldots, L$, and hence, $\|\mathcal{X}_\ell\|_F \leq \|\mathcal{X}_{\ell-1}\|_F \leq \|\mathcal{X}_0\|_F$. From the stability of graph filters in Theorem 1 (with Lemma 2), the difference becomes

$$\|\mathcal{X}_L - \hat{\mathcal{X}}_L\|_F \leq \left(2C\epsilon_s \left(1 + \delta\sqrt{N}\right) + C\kappa\epsilon_\tau + \mathcal{O}(\epsilon^2)\right) \|\mathcal{X}_0\|_F + \left\|\hat{\mathcal{X}}_{L-1} - \mathcal{X}_{L-1}\right\|_F. \tag{83}$$

Substituting (82) in (83) recursively, we obtain

$$\|\mathcal{X}_L - \hat{\mathcal{X}}_L\|_F \leq 2CL\epsilon \left(1 + \delta\sqrt{N}\right) \|\mathcal{X}_0\|_F + CL\kappa\epsilon_\tau\|\mathcal{X}_0\|_F + \mathcal{O}(\epsilon^2). \tag{84}$$

Note that the base case of the recursion is calculated as

$$\|\mathcal{X}_1 - \hat{\mathcal{X}}_1\|_F \leq \left\|\mathbf{H}_1(\mathbf{S}, \mathcal{L}_\tau)\mathcal{X}_0 - \mathbf{H}_1(\hat{\mathbf{S}}, \hat{\mathcal{L}}_\tau)\mathcal{X}_0\right\|_F$$
$$\leq \left\|\mathbf{H}_1(\mathbf{S}, \mathcal{L}_\tau) - \mathbf{H}_1(\hat{\mathbf{S}}, \hat{\mathcal{L}}_\tau)\right\| \|\mathcal{X}_0\|_F \tag{85}$$
$$\leq \left(2C\epsilon_s \left(1 + \delta\sqrt{N}\right) + C\kappa\epsilon_\tau + \mathcal{O}(\epsilon^2)\right) \|\mathcal{X}_0\|_F$$

Eventually, as in (74), it follows that the joint operator distance modulo of the output of ST-GNNs is

$$\|\mathbf{\Phi}(.; \mathcal{H}, \mathbf{S} \circ \mathcal{L}_\tau) - \mathbf{\Phi}(.; \mathcal{H}, \hat{\mathbf{S}} \circ \hat{\mathcal{L}}_\tau)\|_{\mathcal{P}, \mathcal{T}} \leq 2CL\epsilon_s \left(1 + \delta\sqrt{N}\right) + CL\kappa\epsilon_\tau + \mathcal{O}(\epsilon^2), \tag{86}$$

completing the proof.

## F    STABILITY OF MULTIPLE-INPUT MULTIPLE-OUTPUT ST-GNNs

Theorem 2 only considers the case of single-feature layers. In this section, we extend the stability analysis to the case of multiple features. We derive a bound for the difference in the output of ST-GNNs in Corollary 1, where each layer as well as the input signals have multiple features. We let the features at the hidden layers to be the same for simplicity.

**Corollary 1** (ST-GNNs Stability). *Consider the assumptions of Theorem 2 but let $F_\ell = F$ be the number of features per each layer for $1 \leq \ell \leq L-1$. Let $F_0$ and $F_L$ be the number of the features of the input and output signals, respectively. Then,*

$$\|\boldsymbol{\Phi}(.; \mathcal{H}, \mathbf{S} \circ \mathcal{L}_\tau) - \boldsymbol{\Phi}(.; \mathcal{H}, \hat{\mathbf{S}} \circ \hat{\mathcal{L}}_\tau)\|_{\mathcal{P}, \mathcal{T}} \leq$$
$$\sqrt{F_L} \left( F^{L-1} F_0 + \sum_{l=1}^{L-1} F^l \right) \left( 2C\epsilon_s(1 + \delta\sqrt{N}) + C\kappa\epsilon_\tau \right) + \mathcal{O}(\epsilon^2). \tag{87}$$

*Proof.* We aim to find the difference between the output of the $L$th layer before and after adding joint perturbations. For a layer $\ell$ and feature $f$, we can define its output as

$$\mathcal{X}_\ell^f = \sigma \left( \sum_{g=1}^{F} \mathbf{H}_\ell^{fg}(\mathbf{S}, \mathcal{L}_\tau) \mathcal{X}_{\ell-1}^g \right), \tag{88}$$

where $\mathbf{H}_\ell^{fg}$ is a single-input signle-output filter as the one used in Theorem 2. From assumption (A2) of Theorem 2, we have the identity $\|\sigma(\mathbf{x}_2) - \sigma(\mathbf{x}_1)\|_2 \leq \|\mathbf{x}_2 - \mathbf{x}_1\|_2$. We can then bound the output difference for all $1 < \ell < L$:

$$\|\mathcal{X}_\ell^f - \hat{\mathcal{X}}_\ell^f\|_F \overset{(a)}{\leq} \sum_{g=1}^{F} \left\| \mathbf{H}_\ell^{fg}(\mathbf{S}, \mathcal{L}_\tau) \mathcal{X}_{\ell-1}^g - \mathbf{H}_\ell^{fg}(\hat{\mathbf{S}}, \hat{\mathcal{L}}_\tau) \hat{\mathcal{X}}_{\ell-1}^g \right\|_F$$

$$\overset{(b)}{\leq} \sum_{g=1}^{F} \left( \left\| \mathbf{H}_\ell^{fg}(\mathbf{S}, \mathcal{L}_\tau) - \mathbf{H}_\ell^{fg}(\hat{\mathbf{S}}, \hat{\mathcal{L}}_\tau) \right\| \left\| \mathcal{X}_{\ell-1}^g \right\|_F + \left\| \mathbf{H}_\ell^{fg}(\hat{\mathbf{S}}, \hat{\mathcal{L}}_\tau) \right\| \left\| \hat{\mathcal{X}}_{\ell-1}^g - \mathcal{X}_{\ell-1}^g \right\|_F \right),$$

$$\overset{(c)}{\leq} F \left( 2C\epsilon \left( 1 + \delta\sqrt{N} \right) + C\kappa\epsilon_\tau + \mathcal{O}(\epsilon^2) \right) \left\| \mathcal{X}_{\ell-1}^1 \right\|_F + F \left\| \hat{\mathcal{X}}_{\ell-1}^1 - \mathcal{X}_{\ell-1}^1 \right\|_F, \tag{89}$$

where $(a)$ follows from the triangle inequality, $(b)$ from (82), and $(c)$ from Theorem 1. The summation in $(b)$ has equivalent $F$ summands since we assume that all the filters satisfies assumption (A1) in Theorem 2. With applying $\left\| \hat{\mathcal{X}}_{\ell-1}^1 - \mathcal{X}_{\ell-1}^1 \right\|_F$ recursively, we can re-write (89) as

$$\|\mathcal{X}_\ell^f - \hat{\mathcal{X}}_\ell^f\|_F \leq \left( F^{\ell-1} F_0 + \sum_{l=1}^{\ell-1} F^l \right) \left( 2C\epsilon \left( 1 + \delta\sqrt{N} + C\kappa\epsilon_\tau \right) + \mathcal{O}(\epsilon^2) \right). \tag{90}$$

Note that the base case of the recursion is calculated as

$$\|\mathcal{X}_1^f - \hat{\mathcal{X}}_1^f\|_F \leq \sum_{g=1}^{F_0} \left\| \mathbf{H}_1^{fg}(\mathbf{S}, \mathcal{L}_\tau) \mathcal{X}_0^g - \mathbf{H}_1^{fg}(\hat{\mathbf{S}}, \hat{\mathcal{L}}_\tau) \mathcal{X}_0^g \right\|_F$$
$$\leq F_0 \left( 2C\epsilon_s \left( 1 + \delta\sqrt{N} \right) + C\kappa\epsilon_\tau + \mathcal{O}(\epsilon^2) \right) \|\mathcal{X}_0^1\|_F. \tag{91}$$

Finally, we can express the difference between the output at layer $L$. It has multiple features, and therefore, the norm of the difference is expressed as

$$\left\| \mathcal{X}_L - \hat{\mathcal{X}}_L \right\|_F^2 = \sum_{g=1}^{F_L} \left\| \mathcal{X}_L^f - \hat{\mathcal{X}}_L^f \right\|_F^2$$
$$\overset{(a)}{\leq} F_L \left( F^{L-1} F_0 + \sum_{l=1}^{L-1} F^l \right)^2 \left( 2C\epsilon_s \left( 1 + \delta\sqrt{N} \right) + C\kappa\epsilon_\tau + \mathcal{O}(\epsilon^2) \right)^2, \tag{92}$$

where $(a)$ is calculated from (90) for $\ell = L$. Taking the square root yields the inequality in (87), which completes the proof. □

## G   EXTENDED NUMERICAL RESULTS

We consider the problem of decentralized controllers, where we are given a team of $N$ agents, each of which has a position $\mathbf{p}_{i,n} \in \mathbb{R}^2$, a velocity $\mathbf{v}_{i,n} \in \mathbb{R}^2$ and an acceleration $\mathbf{u}_{i,n} \in \mathbb{R}^2$, that are captured at times $nT_s, n \in \mathbb{Z}^+$. The goal is to learn controller actions that allow the agents to move together and complete a specific task. Two tasks are considered in this paper: flocking and unlabeled motion planing. Optimal centralized controllers for the two tasks are derived in the literature. However, centralized controllers require access to the information at all the agents, and therefore, the computation complexity scales fast with the number of agents. With the help of ST-GNNs, we can find decentralized controllers that imitate the centralized solutions according to (22). In this section, we aim to provide a detailed description of the experiments in Section 5 along with further experiments.

**Communication networks.** The underlying graphs represent the communication networks between the agents. Two criteria can be used to assemble the graph. First, each agent is connected to its $M$-nearest neighbors. The graph at a time step $n$ is then represented by a binary graph $\mathcal{G}_n$ such that $(i,j) \in \mathcal{E}_n$ if and only if $j \in \mathcal{N}_{i,n}$ or $i \in \mathcal{N}_{j,n}$, where $\mathcal{N}_{i,n}$ is the set of the $M$-nearest neighbors of the agent $i$. The second is to connect each agent to the neighbors within a communication range $R$. The graph in this case is also represented by a binary graph such that $(i,j) \in \mathcal{E}_n$ if and only if $\|\mathbf{p}_{i,n} - \mathbf{p}_{j,n}\| < R$. In both cases, when the agents move, the graph $\mathcal{G}_n$ changes with $n$.

**ST-GNN Implementation.** As indicated above, the communication networks (i.e., graphs) change with the movement of the agents (i.e., with time). However, ST-GNNs in (7) are designed for fixed graphs. Moreover, ST-GNNs are designed for continuous-time signals, whereas the signals in our experiments are discrete. To turn around these challenges, we implement the FIR space-time graph filter recursively, where at every time step we use the corresponding underlying graph. The output of the filter at a time step $n$ is expressed as

$$\mathbf{y}_n = h_0 \mathbf{x}_n + \sum_{k=1}^{K-1} h_k \left( \prod_{m=1}^{k} \mathbf{S}_{n-m} \right) \mathbf{x}_{n-k}, \tag{93}$$

where $\mathbf{x}_n$ and $\mathbf{S}_n$ are the graph signal and the GSO at time step $n$. The graph diffusion is implemented in (93) with the GSO directly instead of the operator exponential $e^{-\mathbf{S}_n}$. The latter can be expressed as

$$e^{-\mathbf{S}_n} = \sum_{l=0}^{\infty} \frac{(-1)^l}{l!} \mathbf{S}_n^l = \mathbf{I} - \mathbf{S}_n + \mathbf{O}(\mathbf{S}_n), \tag{94}$$

where $\mathbf{O}(\mathbf{S}_n)$ contains the higher-order terms. Equation (94) shows that the GSO is a first-order approximation of $e^{-\mathbf{S}_n}$, and the sign difference is absorbed in the learning parameters $\{h_k\}_{k=0}^{K}$.

### G.1   APPLICATION I: FLOCKING AND NETWORK CONSENSUS

In this application, the agents collaborate to avoid collisions and learn to move according to a reference velocity $\mathbf{r}_n \in \mathbb{R}^2$. The reference velocity is generated randomly as $\mathbf{r}_{n+1} = \mathbf{r}_n + T_s \Delta \mathbf{r}_n$, $\forall n \in \mathbb{Z}^+, n < T$. The initial value $\mathbf{r}_0$ is sampled from a Gaussian distribution and so is $\Delta \mathbf{r}_n$, with zero mean and expected norms $\mathbb{E}[\|\mathbf{r}_0\|]$ and $\mathbb{E}[\|\mathbf{r}_0\|]$, respectively. At each time step $n$, each agent $i$ observes a biased reference velocity $\tilde{\mathbf{r}}_{i,n}$ such that $\tilde{\mathbf{r}}_{i,n} = \mathbf{r}_n + \Delta \tilde{\mathbf{r}}_i$ with $\Delta \tilde{\mathbf{r}}_i$ being white Gaussian with independent and identically-distributed (*i.i.d.*) components. The goal is to learn acceleration actions $\{\mathbf{u}_{i,n}\}_{i,n}$ that allow the agents to form a swarm moving with the same velocity.

**Mobility model.** Given acceleration actions $\{\mathbf{u}_{i,n}\}_{i,n}$, each agent moves according to the equations of motion:

$$\mathbf{v}_{i,n+1} = \mathbf{v}_{i,n} + T_s \mathbf{u}_{i,n}, \quad \mathbf{p}_{i,n+1} = \mathbf{p}_{i,n} + T_s \mathbf{v}_{i,n} + \frac{T_s^2}{2} \mathbf{u}_{i,n}. \tag{95}$$

The initial velocity is assumed to be $\mathbf{v}_{i,0} = \mathbf{r}_0 + \Delta \mathbf{v}$, for all $i$, where $\Delta \mathbf{v}$ is white Gaussian with *i.i.d.* components. In our experiments, we aim to learn accelerations $\{\mathbf{u}_{i,n}\}_{i,n}$ that make $\mathbf{v}_{i,n}$ for all $i$ be as close as possible to $\mathbf{r}_n$ and prevent the position differences $\mathbf{p}_{ij,n} = \mathbf{p}_{i,n} - \mathbf{p}_{j,n}$ from being zero for all $i \neq j$ at any time step $n$. We solve this problem under the constraints $\|\mathbf{u}_{i,n}\|_2 \leq \mu, \forall i, n$.

**Optimal centralized controllers.** The problem described above has an optimal solution, which, for all $i = 1, \ldots, N$ and $n \leq T$, is given as

$$\mathbf{u}_{i,n}^* = \frac{-1}{2T_s} \left( \mathbf{v}_{i,n} - \frac{1}{N} \sum_{j=1}^{N} \tilde{\mathbf{r}}_{j,n} \right) - \frac{1}{2T_s} \sum_{j=1}^{N} \nabla_{\mathbf{p}_{i,n}} \mathcal{C}(\mathbf{p}_{i,n} - \mathbf{p}_{j,n}), \tag{96}$$

if $\|\mathbf{u}_{i,n}^*\|_2 \leq \mu$, and otherwise $\mathbf{u}_{i,n}^* = \mu$. The collision avoidance potential $\mathcal{C}(.)$ is defined as

$$\mathcal{C}(\mathbf{p}_i, \mathbf{p}_j) = \begin{cases} 1/\|\mathbf{p}_{ij}\|_2^2 - \log(\|\mathbf{p}_{ij}\|_2^2), & \|\mathbf{p}_{ij}\|_2 \leq \gamma, \\ 1/\gamma^2 - \log(\gamma^2), & \text{otherwise}, \end{cases} \tag{97}$$

where $\mathbf{p}_{ij} = \mathbf{p}_i - \mathbf{p}_j$, and $\gamma = 1$ (Tanner et al., 2003; Gama et al., 2020b). Note that (96) requires access to data from all the agents, and therefore, this solution is known to be the optimal centralized solution. However, in a decentralized setting, the agents exchange information only within their $K$-neighborhood only. Therefore, our goal is to find acceleration controls that *imitate* the centralized solution in (96).

**Decentralized Controllers.** We compare our results to a decentralized controller that only has access to the information shared within the $K$-neighborhood of the agents. Unlike the centralized policy where all the information is available to the central unit, the agents only have access to the data that they receive from their neighbors. More importantly, the data get delayed through the communication network, and the agents have to make their predictions based on outdated data. Therefore, the estimated accelerations in (96) are approximated by

$$\mathbf{u}_{i,n}^{dec} = \frac{-1}{2T_s} \left( \mathbf{v}_{i,n} - \sum_{k=0}^{K} \frac{1}{K|\mathcal{N}_{i,n}^k|} \sum_{j \in \mathcal{N}_{i,n}^k} \tilde{\mathbf{r}}_{j,(n-k)} \right) - \frac{1}{2T_s} \sum_{k=1}^{K} \sum_{j \in \mathcal{N}_{i,n}^k} \nabla_{\mathbf{p}_{i,n}} \mathcal{C} \left( \mathbf{p}_{i,n} - \mathbf{p}_{j,(n-k)} \right),$$

$$\tag{98}$$

where $|.|$ is the set cardinality, and $\mathcal{N}_{i,n}^k = \{j' \in \mathcal{N}_{j,(n-1)}^{k-1} \mid j \in \mathcal{N}_{i,n}\}$ is the set of the $k$-hop neighbors.

### G.1.1 EXPERIMENT 1: NETWORK CONSENSUS

In this section, we present a detailed description of the training process. Remember that the agents in this experiment are not moving and the graphs are fixed mesh grids. The state variables are the estimated and observed velocities, $\mathbf{v}_{i,n}$ and $\tilde{\mathbf{r}}_{i,n}, \forall i, n$.

**Training.** The dataset is generated according to the mobility model in (95) and (96). The dataset consists of 500 time-varying graph signals $\{\mathbf{X}_m\}_{m=1}^{500}$ that are calculated under optimal centralized policies $\{\mathbf{U}_m^*\}_{m=1}^{500}$. We split the data into 460 examples for training, 20 for validation and 20 for testing. We train a 2-layer ST-GNN on the training data and optimize the mean squared loss using ADAM algorithm with learning rate 0.01 and decaying factors $\beta_1 = 0.9$ and $\beta_2 = 0.999$.[1] We then keep the model with the lowest cost (across the validation data) among 30 epochs while the cost is averaged over the $T$ steps. For a single time step, the cost is calculated as

$$c(\mathbf{u}_n) = \frac{1}{2N} \sum_{i=1}^{N} \left\| \mathbf{v}_{i,n} - \frac{1}{N} \sum_{j=1}^{N} \tilde{\mathbf{r}}_{j,n} \right\|_2^2 + \frac{1}{2N} \sum_{i=1}^{N} \|T_s \mathbf{u}_{i,n}\|_2^2, \ \forall n < T. \tag{99}$$

All the training and simulation parameters are shown in Table 1.

**Execution.** In addition to the 20 test examples generated with the training set, we generate another 20 examples under joint graph and time perturbations for each value of $\epsilon$. The perturbation size of both graph and time topologies is chosen the same. The results shown in Fig. 1 (Middle) suggest that the closer the space-time structures, the closer the outputs of the trained ST-GNN under the two cases are. This shows that an ST-GNN, which is trained with signals defined on one particular graph and sampled at a certain sampling rate, can be generalized to signals with a different underlying topology as long as the two topologies are close.

---

[1]We used the GNN library at `https://github.com/alelab-upenn/graph-neural-networks`

Table 1: Simulation parameters in Experiments #1 and #2.

| parameter | value |
|---|---|
| $\mathbb{E}[\|\mathbf{r}_0\|] = \mathbb{E}[\|\Delta\mathbf{r}_n\|]$ | $1 m/s$ |
| $\mathbb{E}[\|\Delta\tilde{\mathbf{r}}_i\|] = \mathbb{E}[\|\Delta\mathbf{v}\|]$ | $1 m/s$ |
| Initial agent density, $\rho_0$ | $0.5$ agents$/m^2$ |
| Communication range, $R$ | $2\ m$ |
| Maximum acceleration value, $\mu$ | $3\ m/s^2$ |
| Time steps, $T$ | 100 |
| Sampling time, $T_s$ | $0.1\ s$ |
| ST-GNN feature/layer, $F_{0:2}$ | $4, 16, 2$ (#1) and $6, 64, 2$ (#2) |
| Filter taps/layer, $K_{1:2}$ | $4, 1$ |
| Activation function, $\sigma$ | tanh |

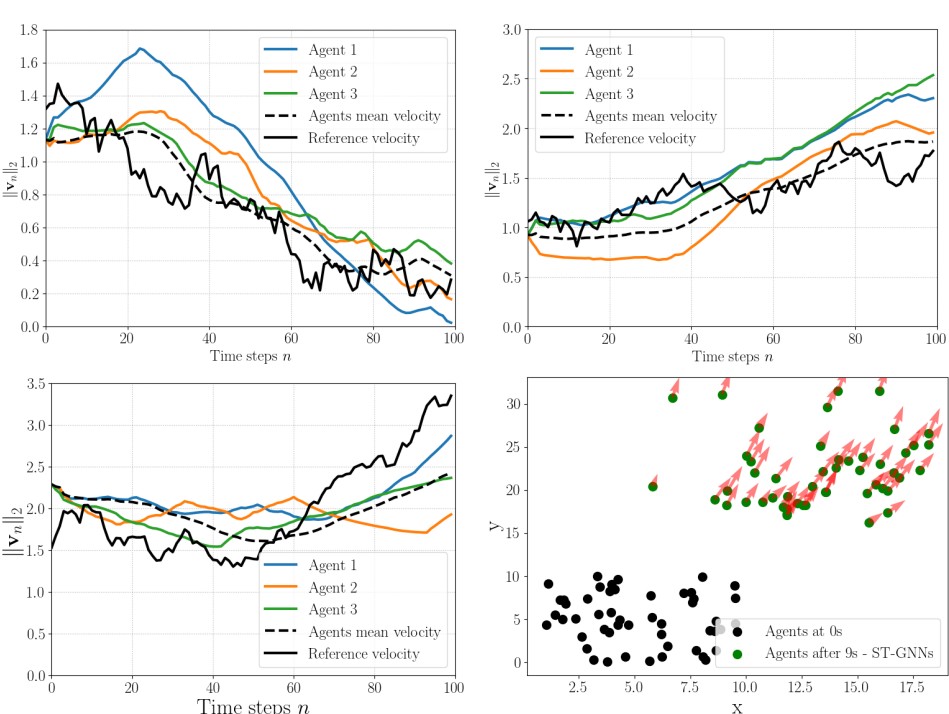

Figure 3: Flocking experiment. (a) (b) and (c) the estimated velocities of some agents in three different examples from the test dataset, and (d) the positions of the agents in (c) at the start of the simulations and after $9s$ (the red arrows represent the agent velocity).

### G.1.2 EXPERIMENT 2: FLOCKING

In this experiment, we have 50 agents that are spread uniformly with initial density $\rho_0$. The agents are allowed to move with their velocities $\{\mathbf{v}_{i,n}\}_{i,n}$. At each time step $n$, the graph $\mathcal{G}_n$ represents their communication network based on a communication range $R$. We train a 2-layer ST-GNN that is implemented in (93), and the training procedure is exactly as in the previous experiment. The dataset consists of 800 training, 100 validation, and 100 test examples. We execute the trained ST-GNNs on signals defined over graphs constructed with the same initial agent density $\rho_0$ and sampled at the same sampling rate $1/T_s$. Fig. 3 depicts three paradigms of the learned velocities compared to the reference velocity. We notice that the agents follow the trend in the reference velocity. The mean and variance of the difference between the agent and reference velocities were shown before in Fig. 1 (Right). Fig. 3d illustrates the positions of the swarm after $9s$ and shows that the swarm moved together with same velocity in the same direction. Fig. 1 (Right) also shows that the proposed architecture outperforms the decentralized policy in (98). The difference between the estimated and reference velocities is lower under ST-GNNs than the difference under the decentralized policy.

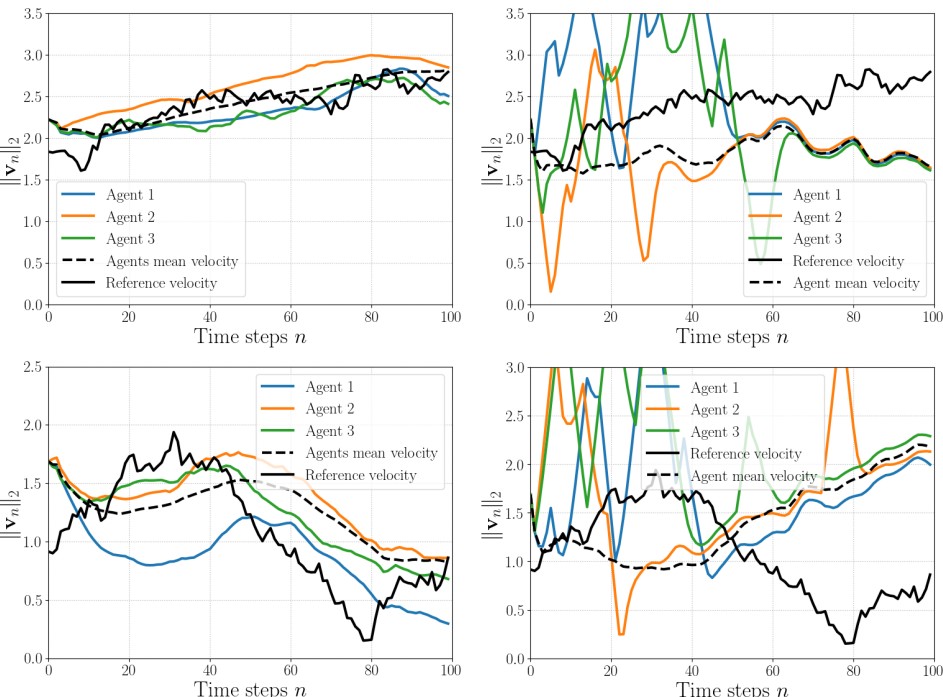

Figure 4: The estimated velocities of some agents in two different examples from the test dataset using ST-GNNs (left) and a decentralized controller (right).

Examples of the test dataset are shown in Fig. 4. It is clear that ST-GNNs help the agents to mitigate the delays in the received information and learn accelerations that follow the reference velocity.

Graph and time perturbations appear in real applications as a change in the underlying graph-time topology. In the following, we provide extended experiments to show the effect of perturbations. In particular, we repeat the same experiment either with graphs constructed with different agent densities or under different values of sampling time $T_s$.

**Experiment 2.I.** In this experiment, we execute the trained ST-GNN on graphs constructed under different initial agent densities (i.e., a source of graph perturbations). We execute the trained ST-GNNs on 50 signals generated with a different initial agent density. We repeat the experiments under densities $2, \frac{1}{2}, \frac{1}{8}, \frac{1}{32}, \frac{1}{128}, \frac{1}{512}$ agents/$m^2$, respectively, and plot the relative costs after $10s$ in Figure 5 (Left). The cost represents the mean difference between the agent velocity and the mean of the observed velocities, which is calculated as

$$\text{cost} = \frac{1}{2N} \sum_{i=1}^{N} \left\| \mathbf{v}_{i,T} - \frac{1}{N} \sum_{j=1}^{N} \tilde{\mathbf{r}}_{j,T} \right\|_2^2. \tag{100}$$

The relative cost is then calculated as a relative deviation from the cost under a density of 2 agents/$m^2$. Figure 5 (Left) shows that for the small changes in the density $\Delta\rho$ (e.g., from the original density 2 to 0.5 agents/$m^2$), the relative cost remains small. However, the higher values of $\Delta\rho$ result in higher relative costs, i.e., degradation in the performance. Note that using smaller agent densities (i.e., higher $\Delta\rho$) at the same communication range $R$ results in sparser graphs. Therefore, their distances to the graphs generated at the original density (2 agents/$s$) increases leading to the aforementioned performance degradation.

**Experiment 2.II.** In this experiment, we execute the trained ST-GNN on signals sampled at different sampling rates (i.e., a source of time perturbations). For several values of $\Delta T_s$, we replace the sampling time $T_s$ with $T_s + \Delta T_s$ and calculate the relative costs as in Experiment 2. Figure 5 (Right) shows that the smaller the value of $\Delta T_s$, the smaller the relative cost is. This matches our theoretical results.

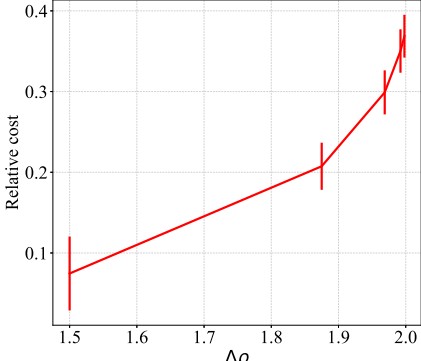 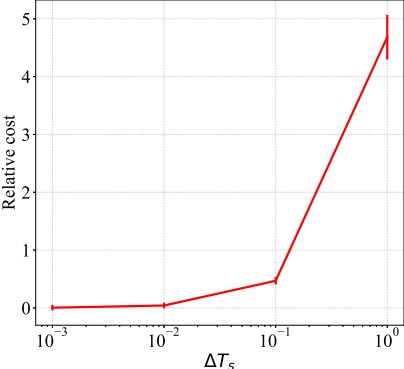

Figure 5: Relative cost after $10s$ calculated over test datsets that have encountered either graph perturbations resulted from changing the agent density $\rho_0$ (Left), or time perturbations resulted from changing the sampling time $T_s$ (Right).

Table 2: Simulation parameters in Section G.2

| parameter | value |
|---|---|
| No. of agents, $N$ | 12 |
| Neighborhood size, $M$ | 5 |
| Initial minimum distance, $d$ | $1.5\ m$ |
| Initial velocity, $\mathbf{v}_{i,0}$ | $4\ m/s$ |
| Maximum acceleration value, $\mu$ | $5\ m/s^2$ |
| Time steps, $T$ | 30 |
| Sampling time, $T_s$ | $0.1\ s$ |
| ST-GNN feature/layer, $F_{0:2}$ | $34, 64, 2$ |
| Filter taps/layer, $K_{1:2}$ | $3, 1$ |
| Activation function, $\sigma$ | tanh |

## G.2 APPLICATION II: UNLABELED MOTION PLANNING

In this problem, we aim to assign $N$ unlabeled agents to $N$ target goals, $\{\mathbf{g}_j \in \mathbb{R}^2\}_{j=1}^N$ through planning their free-collision trajectories. The term *unlabeled* implies that the assignment is not pre-determined and it is executed online. At the start of the experiment, the agents are spread with minimum inter-agent distance $d$ and so do the goal targets. A centralized solution to this problem was introduced in (Turpin et al., 2014), which we refer to as the CAPT solution. This solution gives the agent trajectories, $\{\mathbf{Q}_i \in \mathbb{R}^{2 \times T}\}_{i=1}^N$, while the agent velocities and accelerations can be calculated as

$$\mathbf{v}_{i,n} = (\mathbf{p}_{i,n+1} - \mathbf{p}_{i,n})/T_s, \quad \mathbf{u}_{i,n}^* = (\mathbf{p}_{i,n+1} - \mathbf{p}_{i,n})/T_s^2. \tag{101}$$

Note that $\mathbf{p}_{i,n}$ is the $n$-th column of matrix $\mathbf{Q}_i$. We use the optimal accelerations in (101) to learn an ST-GNN parameterization that predicts the agent accelerations $\{\mathbf{u}_{i,n}\}_{i,n}$ according to (22). The input $\mathbf{X}_m$ consists of $6M + 4$ state variables for each agent, which are the agent position $\{\mathbf{p}_{i,n}\}_n$ and velocity $\{\mathbf{v}_{i,n}\}_n$, the position of the nearest $M$ neighbors $\{\mathbf{P}_{i,n} \in \mathbb{R}^{M \times 2} \mid [\mathbf{P}_{i,n}]_j = \mathbf{p}_{j,n} \ \forall j \in \mathcal{N}_{i,n}\}_n$ along with their velocities, and the position of the nearest $M$ target goals. The CAPT accelerations and the corresponding state variables constitutes together one pair in the dataset.

The dataset consists of 55000 training examples and 125 validation examples. We train an ST-GNN on the training data and optimize the mean squared loss using ADAM algorithm with learning rate 0.0005 and decaying factors $\beta_1 = 0.9$ and $\beta_2 = 0.999$. We then keep the model with the lowest mean distance between the agent final position and its desired target among 60 epochs. The other training parameters are shown in Table 2. The trained ST-GNN is then executed on a test dataset that consists of 1000 examples. The ST-GNN output for each example is the estimated accelerations, and the planned trajectories are then calculated using (95). One example was shown in Fig. 2 (Left), which depicts free-collision trajectories.

Similar to the flocking experiment, we test the trained ST-GNNs with different graph-time topologies, generated with either different neighborhood sizes or different sampling times. Figures 2 (Middle) and (Right) uphold the conviction that the ST-GNN output difference increases with the distance between the underlying topologies.

