# OpenReview forum: "Space-Time Graph Neural Networks"
_ICLR.cc/2022/Conference — ICLR 2022 Poster_

### Official Review · Reviewer_aJtV · 2021-10-24

**Correctness:** 3
**Technical Novelty And Significance:** 3
**Empirical Novelty And Significance:** 2
**Recommendation:** 5
**Confidence:** 3

**Main Review:**

Strengths:
1.  Solid theoretical support.
2.  Jointly process graph and time domains in a single convolution operation.
3.  Stable to small variations in graphs and sampling rates.

Weakness
1. Lack of comparison with other methods.
2. Lack of time complexity analysis.
3. The setting of hyper-parameters is not specified.


**Summary Of The Paper:**

This paper introduces a space-time convolution operator that is robust to graph perturbations and small variations of sampling rates.

**Summary Of The Review:**

Although this paper theoretically proves the stability of the proposed convolution operator, it does not provide experimental evidence about the superiority of the proposed method against other baseline methods. Given this paper is not the first to study time-varying graph signals, I think baseline comparison is necessary. Based on the above consideration, I think this paper is not ready for acceptance.

---

> ### Author Response · Authors · 2021-11-23
> **Responses to Reviewer aJtV**
>
> We would like to thank the reviewer for their valuable comments.
>
> 1) In comment 1.2 and 2.1 we explain the uniqueness of the proposed architecture, which makes it a perfect fit for physical networks and decentralized applications. Comparing with the aforementioned architectures is impractical (as explained previously). However, there exists an algorithm in the decentralized controllers field that we add as a baseline in the revised version of the paper.
>
> 2) The complexity analysis of our proposed method is the complexity analysis of the classical convolutional neural network architecture, and therefore, it was omitted from the discussion.
>
> 3) We refer the reviewer to Appendix G where all hyper-parameters are specified.

---

### Official Review · Reviewer_G6p1 · 2021-11-04

**Correctness:** 4
**Technical Novelty And Significance:** 3
**Empirical Novelty And Significance:** 2
**Recommendation:** 5
**Confidence:** 3

**Main Review:**

Strengths:
1.	The paper is well-written and, despite heavily populated with math formulas and notation, I find it enjoyable and relatively easy to follow it through.
2.	The proposed ST-GNN is principled, natural and neat.
3.	I like the analysis of ST-GNN’s stability to perturbations. I find it new and insightful, and especially interesting to read, though I am not an expert in this so I may be biased as well.

Weaknesses:
1.	I am not sure if the experiments would be very appealing to the deep learning community because they do not compare with the many latest temporal network SOTAs, and the two tasks seem to mostly target audience from robotics and control. Technically speaking, it mainly uses temporal network dataset whose features have strong continuous-time dynamics, and only for node-level prediction task). To appeal to more audience from the ICLR community, I would suggest that the experiments be significantly extended by referencing and comparing with the following papers in recent years: [1, 2, 3, 4, 5]. Alternatively, the authors could be more explicit in restricting ST-GNN's application domain.
2.	Despite being well-principled and neat, the proposed space-time convolution (Equation 7) does not seem quite new to me. It seems essentially just a temporal convolution followed by graph diffusion which is not totally new ([6] for example adopted a very similar idea).

[1] Inductive Representation Learning on Temporal Graphs
[2] Inductive Representation Learning in Temporal Networks via Causal Anonymous Walks
[3] EvolveGCN: Evolving Graph Convolutional Networks for Dynamic Graphs
[4] Variational Graph Recurrent Neural Networks
[5] JODIE: Predicting Dynamic Embedding Trajectory in Temporal Interaction Networks
[6] TEDIC: Neural Modeling of Behavioral Patterns in Dynamic Social Interaction Networks


**Summary Of The Paper:**

This paper introduces a new spatio-temporal Graph Neural Network, ST-GNN, for making predictions on temporal network. Its proposed space-time convolution operator is a composition of temporal convolution and graph diffusion. The paper further proves that under practical conditions their ST-GNN with Integral Lipschitz filters is stale to small perturbation in both time and graph domain.

**Summary Of The Review:**

This is a well-written paper with a neat space-time convolution model and important theoretical contributions in analyzing its stability. However, as significant as its strengths are, its weaknesses are also concerning, which mainly involves two aspects: 1. the architecture of ST-GNN does not seem novel and may suffer from varying graph size; 2. it certain could have empirical studies done on more datasets and have more baselines compared.  Overall, I would be very happy to improve the score if we could see more extensive experiment with more baselines and positive results.

---

> ### Author Response · Authors · 2021-11-23
> **Responses to Reviewer G6p1**
>
> We would like to thank the reviewer for enjoying our work and their insightful comments. Note that in our response and the submitted paper, graph signals correspond to signals or data that are available at each node. In literature, graph signals are also referred as node attributes.
>
> 1) We agree with the reviewer that the range of applications that are well suited for ST-GNNs should be better explained and thus we have added a discussion in the revised version. However, we believe that the considered applications are very appealing to the ML community. Whereas standard prediction tasks are vastly studied and a plethora of architectures have been proposed, physical network applications are still under-investigated. ST-GNNs try to close this gap and are designed to process physical networks as sensor networks, multi-agent systems, transaction systems, to name a few. In physical networks, data naturally get delayed based on the physical constraints, surrounding environments or other reasons. As a result agents are asked to make decisions and predictions based on out-dated information. Furthermore, decision making is naturally decentralized in a highly-dynamic environment and each agent makes a decision based on the data communicated with the other agents.
>
>    The current SOTA in time-varying GNNs does not take into account these delays and the predictions are done based on the data of all nodes up to the present point. They, therefore, make finer predictions due to their access to broader and updated data, but they are inapplicable since they contradict the physical restrictions of the problem. ST-GNNs, on the other hand respect and exploit these delays, and below we highlight the difference between ST-GNNs and the architectures in [1-5].
>
>    Specifically, the work in [5] is proposed to learn representations of users and items in temporal interaction networks. The architecture in [5] does not utilize graph signals or attributes and is closer to architectures used for knowledge graphs. ST-GNNs, on the other hand, are designed for time-varying graph signals, therefore the two architectures cannot be compared. Furthermore, the works in  [1], [2] do consider time-varying graphs but not time-varying graph signals, i.e., the signals (attributes) at each node remain constant over time. Therefore, [1] and [2] are not applicable to the considered tasks where both the graph and the graph signal are time-varying. The works in [3, 4] are the closest to our work since they are designed for temporal graphs and time-varying graph signals. The difference lies in the use of the time shift operator (TSO) that our work proposes. To see the effect of the TSO let's focus on equation (93). For node i the output $y_n(i)$ is $h_o x_n(i) + \sum h_k \Pi S_{n-m} x_{n-k} $. So, at time step n the output $y_n(i)$ at node i, is a function of the signal of node i at time n, and the signals of the k-hop neighbors of node i at time n-k. The works in [3, 4], however, utilize filters where the output $y_n(i)$ is a function of the signal of node i and the signals of the 1-hop neighbors of i at time n. Since $S_{n} x_{n}$ diffusions are involved in each layer, the signals of k-hop neighbors of i are required at time n. In other words, the use of the TSO results in diffusions of $\Pi S_{n-m} x_{n-k}$ type, that respect the time delay required to propagate information through the network. As a result in applications where the global information of the graph signals is not available at each node, as flocking and unlabelled motion planning, the works in [3, 4] cannot be applied. To summarize [3, 4] are designed for centralized applications where global information is required at a node level, whereas the proposed ST-GNN respects the time delays in the network and is a better fit for decentralized tasks. (See also comment 1.2)
>
>     In the revised version of the paper we elucidate the range of applications that ST-GNNs are well suited for, and mention the differences with the architectures of [3, 4]. Furthermore, we add comparisons with decentralized controllers which apply decentralized policies and can be fairly compared with our method.
>
>
> 2) The reviewer is right that space-time convolution involves two operations, temporal convolution (shift) and graph diffusion. The work in [6] also utilizes temporal convolution and  graph diffusion. However, there is a significant difference between the two works. Our proposed-space time convolution is defined by the composition of graph and time shift operator which jointly applies graph and time convolution to the graph signal. The work in [6] first applies a graph diffusion and then a temporal convolution, thus the operation is sequential and not joint. As a result, the operation in [6] is more restrictive and requires each node to posses global information about the node attributes at every time instance. This happens because graph diffusion precedes time convolution.

---

### Official Review · Reviewer_u6ex · 2021-11-06

**Correctness:** 4
**Technical Novelty And Significance:** 3
**Empirical Novelty And Significance:** 3
**Recommendation:** 8
**Confidence:** 3

**Main Review:**

-- Strengths:

1. The paper is very well-written and easy to follow. It was a pleasure to read the novel theoretical material presented in this paper.

2. The paper proposes a new architecture based on their new operators and implementing the finite impulse response (FIR) filter recursively.

3. The theoretical material in the paper is quite thorough and very detailed. In the appendix F, the authors also extend their stability analysis in the main paper to not only consider single-layer features but also extend their analysis to multiple features, which makes the theoretical contributions even stronger.

-- Weaknesses:

1. The experiments in this paper are done on synthetically generated datasets for two applications of flocking and unlabelled motion planning. I would have liked to see experiments done on more real-world mobility datasets to better understand the applicability of the proposed method. It is evident from the in-depth theoretical analysis that the author’s primary focus is to propose a novel architecture that comes with robustness guarantees. But, I feel that some experiments on real-world datasets would further strengthen this paper’s contributions.

2. Can this method also be applied to “standard prediction tasks” on time-varying graphs? If so, could you please elaborate on this and how it would compare to other existing networks for time-varying graphs, e.g. graphon neural networks?

3. If ST-GNN can also be used on standard prediction tasks and do a better job than other GNNs, I was wondering how it would handle the cases where we have heterophilous vs. homophilous time-varying networks? This would also be interesting to investigate.


**Summary Of The Paper:**

This paper proposes a new deep learning architecture called ST-GNN that learns representations on graphs that evolve over time. Their work focuses on developing an interesting time-varying convolutional architecture, which exploits the graph-time underlying structure of the signals, processing across both the graph and time domains. The authors introduce new graph shift operators (GSO), time shift operators (TSO), followed by a linear combination of those which results in a space-time shift operator termed STSO. The advantage of this paper is that it can handle continuous-time graph signals and the stability of this proposed architecture is studied theoretically (Section 4) with a significant result outlined in Proposition 1 of the paper where the difference between the space-time graph filters of the original graph and the perturbed graph are upper bounded by the order of error introduced.


**Summary Of The Review:**

This paper proposes a novel architecture based on graph-shift, time-shift and space-time shift operators (also proposed by the authors in the paper) to handle time-varying graphs. The theoretical analysis of this paper is very thorough and clearly elucidates the stability of their NN architecture.

---

> ### Author Response · Authors · 2021-11-23
> **Responses to Reviewer u6ex**
>
> We would like to thank the reviewer for appreciating our work and providing essential comments. Note that in our response and the submitted paper, graph signals correspond to signals or data that are available at each node. In literature, graph signals are also referred as node attributes.
>
> 1) We agree with the reviewer that real-world datasets would further strengthen the value of our contributions. However, datasets with high fidelity in the temporal domain of the signal are not publicly available or easily accessible. Therefore, in our experiments, we generate data in a practical manner that also respect the physics laws. Note that this is a common practice in the literature that studies the considered applications.
>
> 2) The proposed architecture can be used for standard prediction tasks, e.g., node classification and link prediction, when time-varying graph signals are involved. However, ST-GNNs are more applicable to physical-network tasks were time delays are present and predictions are made in a decentralized manner. The main advantage of ST-GNNs compared to existing methods is that they exploit the time-varying structure of the graph signals along with the graph structure.  To be more precise, lets focus on equation (93). At time step n each node i processes the signal of node i that becomes available at time n, along with the signals of the k-hop neighbors of node i that become available at time n-k. In other words, the proposed ST-GNN employs time-delayed information to reconcile for the fact that nodes do not have access to the latest global graph signal information at time n. As a result, ST-GNNS are tailored to decentralized physical-network applications, as those considered in the paper, whereas classical GNN approaches are more natural to centralized tasks where all nodes have access to global information at every-time step. (See also the response to 2.1)
>
> 3) An heterophilous network is a good example where global attribute information might not be available to each node and information can only be shared to neighboring nodes. Therefore, applying an ST-GNN to such networks is a good idea. However, homophilous networks are also involved in physical systems and decentralized tasks. Thus, the applicability of ST-GNNs is affected more by the presence of time delays in the network and the application (centralized vs decentralized), rather than homophilous vs heterophilous networks.

---

### Decision · Program_Chairs · 2022-01-20

**Decision:**

Accept (Poster)

**Comment:**

This paper proposes a new  time-varying convolutional architecture (ST-GNN) for dynamic graphs. The reviewers were positive about the presentation and detailed theory, especially on the stability analysis. The shared criticism was on experimental validation synthetic datasets that the reviewers did not find appealing. The AC believes that while the lacking validation concerns are legit, there is a lack of sophisticated dynamic graph benchmarks in the community yet, so the authors did their best effort to test their method. We thus recommend to accept the paper.